# Deriving stratospheric age of air spectra using an idealized set of chemically active trace gases

Marius Hauck[1], Frauke Fritsch[2,3], Hella Garny[2,3] and Andreas Engel[1]

[1]Institute for Atmospheric and Environmental Sciences, Goethe University Frankfurt am Main, Frankfurt am Main, Germany
[2]Deutsches Zentrum für Luft- und Raumfahrt (DLR), Institut für Physik der Atmosphäre, Oberpfaffenhofen, Germany
[3]Ludwig Maximilians University of Munich, Meteorological Institute Munich, Munich, Germany

*Correspondence to*: Marius Hauck (hauck@iau.uni-frankfurt.de)

**Abstract.** Analysis of stratospheric transport from an observational point of view is frequently realized by evaluation of mean age of air values from long-lived trace gases. However, this provides more insight into general transport strength and less into its mechanism. Deriving complete transit time distributions (age spectra) is desirable, but their deduction from direct measurements is difficult and so far primarily based on model work. This paper introduces a modified version of an inverse method to infer age spectra from mixing ratios of short-lived trace gases and investigates its basic principle in an idealized model simulation. For a full description of transport seasonality the method includes an imposed seasonal cycle to gain multimodal spectra. An EMAC model simulation is utilized for a general proof of concept of the method and features an idealized dataset of 40 radioactive trace gases with different chemical lifetimes as well as 40 chemically inert pulsed trace gases to calculate pulse age spectra. It is assessed whether the modified inverse method in combination with the seasonal cycle can provide matching age spectra when chemistry is well-known. Annual and seasonal mean inverse spectra are compared to pulse spectra including first and second moments as well as the ratio between them to assess the performance on these time scales. Results indicate that the modified inverse age spectra match the annual and seasonal pulse age spectra well on global scale beyond 1.5 years mean age of air. The imposed seasonal cycle emerges as a reliable tool to include transport seasonality in the age spectra. Below 1.5 years mean age of air, tropospheric influence intensifies and breaks the assumption of single entry through the tropical tropopause, leading to inaccurate spectra in particular in the northern hemisphere. The imposed seasonal cycle wrongly prescribes seasonal entry in this lower region and does not lead to a better agreement between inverse and pulse age spectra without further improvement. Tests with focus on future application to observational data imply that subsets of trace gases with 5 to 10 species are sufficient to derive well-matching age spectra. These subsets can also compensate an average uncertainty of up to ±20 % in the knowledge of chemical lifetime if a deviation of circa ±10 % in modal age and amplitude of the resulting spectra is tolerated.

## 1 Introduction

Stratospheric meridional circulation, referred to as Brewer-Dobson circulation (BDC), is a key process for the comprehension of air mass transport throughout the atmosphere. The spatial distributions and atmospheric lifetimes of various greenhouse gases and ozone depleting substances, such as halocarbons, are strongly influenced by this large-scale motion (Butchart and Scaife, 2001; Solomon et al., 2010). Therefore, the BDC affects not only the chemical composition of the stratosphere, but also the radiative budget of the complete atmosphere. The BDC is a combination of a residual mean circulation with net mass flux and eddy-induced isentropic bidirectional mixing (Plumb, 2002; Shepherd, 2007; Butchart, 2014). Air is transported mainly through the tropical tropopause and then advected to higher latitudes, where it eventually descends. Primary drivers are tropospheric planetary- and synoptic-scale Rossby waves that propagate upward and transfer their momentum by breaking in the extra-tropical middle stratosphere (Haynes et al., 1991; Holton et al., 1995). At the same time, this wave drag induces stirring processes that are especially enhanced in this "surf zone" (McIntyre and Palmer, 1984). It has been shown that the tropical upward mass flux has a distinct seasonal cycle with a maximum during northern hemispheric (NH) winter time, when wave excitation is largest (Rosenlof and Holton, 1993; Rosenlof, 1995).

A problem that arises especially regarding observational investigation of the stratospheric circulation is the impossibility of direct measurements of the underlying dynamics, i.e. slow overturning circulation. However, a suited tool for quantification, that can be derived from observations of chemically very long-lived trace gases and directly compared to model results, is the concept of mean age of air (AoA) (Hall and Plumb, 1994; Waugh and Hall, 2002). Mean AoA can be understood as the average period of time that elapsed for an air parcel at any arbitrary location since passing a certain reference point. Usually, the reference is either earth's surface or the tropical tropopause layer. Mean AoA provides not only insight into the current overall strength of the BDC, but also allows for an investigation of temporal changes. If the circulation intensity varies over time, the value of mean AoA will also show this trend, but inversely proportional (Austin and Li, 2006). Different models predict an enhanced stratospheric circulation indicated by a negative trend of mean AoA (Garcia and Randel, 2008; Li et al., 2008; Oman et al., 2009; Shepherd and McLandress, 2011) as response to strengthened wave drag by rising greenhouse gas concentrations. On the other hand, sparse observationally derived mean AoA from balloon-borne $SF_6$ and $CO_2$ data by Engel et al. (2009) show an insignificant positive change between 1975 and 2005 in northern mid-latitude stratosphere above 24 km altitude. Recently, this trend is further affirmed by Engel et al. (2017), where existing data are extended and still show an insignificant trend for the same spatial region. Ray et al. (2014) reexamined these data of $SF_6$ and $CO_2$ using a simplified tropical leaky pipe model and even find a statistically significant positive trend in mean AoA at similar ranges of altitude and latitude as Engel et al. (2009). The analyses of satellite data by Stiller et al. (2012) and Haenel et al. (2015) additionally indicate that the temporal changes of mean AoA exhibit hemispheric asymmetries. Analysis of in situ trace gas measurements of $N_2O$, $O_3$ and model trajectories by Bönisch et al. (2011) suggest an increased tropical upwelling combined with an enhanced transport from the tropical into the extra-tropical lower stratosphere. These results connote that the strength of the BDC is not changing

uniformly. Similar findings are presented by Hegglin et al. (2014) using merged satellite $H_2O$ data. In fact, Birner and Bönisch (2011) propose a separation of the residual mean circulation into two distinct branches conveying air from the tropics into the extra-tropics. The shallow branch is mainly effective in the lower stratosphere, while air in the middle and upper stratosphere is primarily affected by the deep branch. Since more recent studies of models (Okamoto et al., 2011; Oberländer-Hayn et al.,

2015), reanalyses (Monge-Sanz et al., 2013; Abalos et al., 2015) and observations (Ray et al., 2014; Haenel et al., 2015; Engel et al., 2017) still exhibit some inconsistencies about possible future changes in the strength of the circulation pathways, further analyses are necessary. Recent studies of model and satellite observations also suggest that a southward (Stiller et al., 2017) and upward (Oberländer-Hayn et al., 2016) shift of the BDC might be key factors for these inconsistencies.

For an even more thorough analysis of stratospheric transport, the usage of a full transit time distribution is of advantage. The interaction of mean residual transport and bidirectional mixing as well as the influence of shallow and deep branch is expressed in this distribution, which is also referred to as age spectrum (Hall and Plumb, 1994; Waugh and Hall, 2002). At any given point in the stratosphere, the age spectrum denotes the fraction of fluid elements in an air parcel that have had a certain transit time from the reference point to the chosen location. It can be considered a probability density function (PDF) with the mean

AoA being the first moment of this distribution. An important advantage is that changes in the different ranges of transit times of the BDC can be visualized with a comparison of age spectra at different points in time. Variations of the shallow branch influence the age spectrum mostly at short transit times (ca. 1 to 2 years (Birner and Bönisch, 2011)), whereas changes of the deep branch mostly at long transit times (ca. 4 to 5 years (Birner and Bönisch, 2011)). Additionally, the effect of aging by mixing (Garny et al., 2014) on the tail of the spectrum may also be assessed during such an analysis. In model experiments,

the age spectrum can be gained via periodically occurring pulses of a chemically inert trace gas. For mean AoA, either a linearly increasing chemically inert trace gas or the mean of the age spectrum can be applied. In reality, only few very long-lived trace gases exhibit a linear trend in the first approximation and can be utilized to gain mean AoA (Andrews et al., 2001a). When deriving mean AoA from observations of very long-lived trace gases with non-linear increase, however, assumptions about the shape of the age spectrum are required, where mostly the pioneering work of Hall and Plumb (1994) is considered.

The underlying age spectrum for this calculation is only an approximation, that might not be representative in all cases and could bias the inferred mean AoA. A directly inferred complete age spectrum may constitute an improvement and could give further insight into transport processes. This information might be extracted from mixing ratios of long- and short-lived trace gases in an air parcel as for such species residual transport, mixing and chemical depletion are inevitably associated with each other. The state of depletion of these gases provides an estimate of the elapsed transit time since passing the reference point.

In addition, trace gases with varying chemical depletion are only sensitive to certain transport pathways and thus contribute mainly to the age spectrum for transit time ranges up to their respective chemical lifetime (Schoeberl et al., 2000; Schoeberl et al., 2005). The complete age spectrum may then theoretically be derived as combination of those pieces of information provided that a sufficient number of distinct mixing ratios are known. Since the amount of air parcels with transit times larger than ten years is likely to be low even in the uppermost part of the stratosphere, trace gases with lifetimes up to ten years

should be sufficient to retrieve a meaningful age spectrum. That makes short-lived trace gases a suitable tool as a variety of species with diverse lifetimes were frequently measured during past airborne research campaigns. Unfortunately, to the best of our knowledge, there are only few publications about possible techniques to finally convert the information of the mixing ratios of short-lived trace gases into stratospheric age spectra (Schoeberl et al., 2005; Ehhalt et al., 2007) and none that include seasonality in transport. When analyzing seasonal variation in stratospheric dynamics by inferring age spectra from observations, though, a proper consideration of the seasonal cycle is also required to achieve reliable results.

This paper presents an application and evaluation of a modified version of the method by Schoeberl et al. (2005) (Schoeberl's method) with a reduced set of fit parameters and an imposed seasonal cycle to account for seasonality in stratospheric transport. The modified technique is applied as a proof of concept to an idealized simulation of the ECHAM/MESSy Atmospheric Chemistry (EMAC) model (Jöckel et al., 2006; Jöckel et al., 2010). Section 2 provides insight into the method and the model simulation. In Sect. 3, resulting age spectra and related quantities are analyzed and assessed with respect to future application to observational data. Finally, there is a summary, conclusion and outlook in Sect. 4.

## 2 Methodology

### 2.1 Age spectra derivation

A frequent basis for the estimation of age spectra, that is also utilized in Schoeberl et al. (2005) and Ehhalt et al. (2007), is the mathematical description of a mixing ratio $\chi(\vec{x}, t)$ for any substance with a stratospheric sink at an arbitrary location $\vec{x}$ at any time $t$ in the stratosphere given by the following equation

$$\chi(\vec{x}, t) = \int_0^\infty \chi(\overrightarrow{x_0}, t - t') \cdot e^{-\frac{t'}{\tau(t')}} \cdot G(\vec{x}, t, t') \cdot dt'. \tag{1}$$

Here, $t'$ is the transit time through the stratosphere, $\chi(\overrightarrow{x_0}, t - t')$ the mixing ratio time series at the tropical tropopause (entry mixing ratio), $\tau(t')$ the chemical lifetime and $G(\vec{x}, t, t')$ the age spectrum. Usually, a PDF is integrated from $-\infty$ to $\infty$, but since negative transit times are physically undefined for our analysis, 0 is taken instead. To simplify the derivation, a constant annual mean entry mixing ratio $\overline{\chi_0} = \overline{\chi(\overrightarrow{x_0}, t - t')}$ is assumed. This keeps the balance between a physically accurate description and practicability for observational studies, since time series of short-lived species at the tropical tropopause are hard to attain. Eq. ( 1 ) then becomes

$$\chi(\vec{x}, t) = \overline{\chi_0} \cdot \int_0^\infty e^{-\frac{t'}{\tau(t')}} \cdot G(\vec{x}, t, t') \cdot dt'. \tag{2}$$

One of the first approaches to obtain the age spectrum from such an equation in an unidimensional case without chemical loss was made by Hall and Plumb (1994). In their study, the vertical diffusion equation with a constant diffusion coefficient was used to calculate the distribution as a Green's function in response to a Dirac delta distribution at the tropical tropopause. The result is an inverse Gaussian PDF that describes transport reasonably well. Their results show moreover that the ratio of the second and first moment of the age spectrum is rather constant in certain stratospheric regions, a relationship that is still used for the observational estimation of mean AoA from $SF_6$ or $CO_2$. Mathematically, the age spectrum in Hall and Plumb (1994) is described as follows

$$G(z,t') = \frac{z}{2\sqrt{\pi K t'^3}} \cdot e^{\left(\frac{z}{2H} - \frac{Kt'}{4H^2} - \frac{z^2}{4Kt'}\right)} \tag{3}$$

with $K$ as constant diffusion coefficient, $z$ as vertical coordinate and $H$ as scale height of the air density $\varrho$. Latter can either be gained from an altitude-density fit ($\varrho = \varrho_0 \cdot e^{-\frac{z}{H}}$) or approximated by a constant value. The first moment $\Gamma$ (i.e. mean) and the central second moment $\Delta^2$ (i.e. variance) are then defined as (Hall and Plumb, 1994)

$$\Gamma(z) = \int_0^\infty G(z,t') \cdot t' \cdot dt' \tag{4}$$

$$\Delta^2(z) = \frac{1}{2} \cdot \int_0^\infty G(z,t') \cdot (t' - \Gamma(z))^2 \cdot dt'. \tag{5}$$

Transfer of Eq. ( 3 ) to complex stratospheric transport is quite difficult as Hall and Plumb (1994) state that their inverse Gaussian solution for one-dimensional diffusion is restricted to long-lived trace gases (Holton, 1986; Plumb and Ko, 1992) in a setup with only annual mean transport. In such cases, the diffusion coefficient $K$ has to be considered a measure of net vertical advection and mixing rather than simply small-scale diffusion. Their age spectra therefore do not incorporate any seasonal or inter-annual variability in transport. Model studies of annual and seasonal age spectra (Reithmeier et al., 2008; Li et al., 2012a; Li et al., 2012b; Ploeger and Birner, 2016) have shown that age spectra exhibit multiple modes representing seasonal fluctuations of stratospheric transport (see Sect. 2.2). However, even though the inverse Gaussian solution of Hall and Plumb (1994) does not include multiple peaks intrinsically and might not provide a perfect solution at any point in the stratosphere, it can certainly provide a robust approximation of the general (smoothed) shape of these multimodal spectra in models (see e.g. Fig. 3 in Li et al. (2012a), Fig. 7 in Li et al. (2012b) or Fig. 5 in Ploeger and Birner (2016)). A prescribed age spectrum shape, even if only a fit, is indeed valuable for observational studies with very limited data. If the general shape is well constrained a priori, all information from trace gases measurements will solely be used to tune the spectrum until it is as close as possible to real transport. Approaches of this kind require less data than cases with a fully unconstrained shape, where

more information about transport processes is needed. Together with the relatively low amount of free parameters of the inverse Gaussian distribution in Hall and Plumb (1994), these are important factors why Eq. ( 3 ) has been the basis of many studies focusing on the retrieval of age spectra from observational data.

Ehhalt et al. (2007) used the formalism of Hall and Plumb (1994) to derive a method, where mixing ratios of short-lived trace gases are correlated against those of very long-lived ones. This correlation is identical to a vertical profile in a trace-gas-based coordinate system. Technically, the method can also be applied using other vertical coordinates such as altitude, pressure or potential temperature as long as there is an exponential decay in the vertical space. In order for this to work, a height-independent chemical lifetime has to be assumed as Eq. ( 2 ) only then turns into a Laplace transform of the age spectrum and

makes for a constant vertical diffusion coefficient. This is quite an elegant approach, since it allows to find an analytical solution of the equation. The study concluded that the resulting age spectra from data between 30° N and 35° N in the lowermost stratosphere are most probably valid for transit times up to one year. Application of Ehhalt's method is restricted to this very narrow region around the subtropical and mid-latitude tropopause. Complete global stratospheric transport, though, cannot be described by a height-independent diffusion coefficient and constant chemical depletion. Schoeberl et al. (2000) had

already proposed a generalized solution to this problem by discretizing Eq. ( 2 ) for long-lived trace gases with chemistry along an average Lagrangian path and solving the corresponding system of linear equations. This strategy does not require a prescribed age spectrum, but our tests with the method show that the matrix that has to be inverted is close to singular and does not lead to numerically stable solutions even considering appropriate solvers. In a very recent study by Podglajen and Ploeger (2019), this method is modified leading to a stable matrix inversion and promising results. Schoeberl et al. (2005) took a

different approach by allowing both transit-time-dependent lifetimes and a vertically varying diffusion coefficient in the age spectrum of Hall and Plumb (1994). They insert Eq. ( 3 ) into Eq. ( 2 ), replace $t'$ by $(t' - t_{off})$ and perform a multiparameter least squares fit on $z$, $K$ and $t_{off}$ to find a matching age spectrum by minimizing the sum of squared deviations between the observed and inverted trace gas mixing ratio in their model. The temporal offset $t_{off}$ is introduced to incorporate the effect that diffusion does not show an instantaneous response to advected trace gases. In comparison to Ehhalt's method, Schoeberl's

method provides major advantages as it is still straightforward to use but with a less restrictive approximation of ongoing physical mechanisms. It holds a much larger area of applicability in the stratosphere. A similar approach for long-lived trace gases (deseasonalized $CO_2$) has been applied by Andrews et al. (1999) with reasonable results.

There are, however, two points concerning the method's formulation that are worthy of discussion. First, the vertical coordinate

$z$ should be considered as measure of position in the atmosphere to ensure that the result represents the age spectrum exactly at the given location. We propose to exclude it from the fit so that all information about transport is inverted into the time- and space-dependent diffusion coefficient $K(\vec{x}, t)$. Technically, calling $K(\vec{x}, t)$ a diffusion coefficient is imprecise, since it includes information about all occurring transport processes and is not depending on a specific substance. It provides an

undirected measure of transport strength and will therefore be generally referred to as transport parameter. Second, the formulation of the age spectrum by Schoeberl et al. (2005) results in a complex age spectrum for $t' < t_{off}$ as negative values appear in the square root. Chhikara and Folks (1989) demonstrate in their Fig. 2.1 and 2.2 that an inverse Gaussian distribution in its most general form can include an offset when varying only shape parameter and mean. Reasonable age spectra might

still be derived when neglecting an explicit temporal offset in the formulation. With this reflection, the necessary fit parameters reduce to only the diffusion coefficient, simplifying the numerical implementation of the method considerably. This requires also that the tail of the spectrum is considered on a reasonably long time scale to regard the mathematical necessity of integrating from 0 to infinity (see Appx. A for numerical details). Additionally, a single parameter fit is by definition less prone to run in a local instead of a global minimum when optimizing its parameter. Applying those simplifications to Eq. ( 2 ) and

Eq. ( 3 )  leads to

$$
\frac{\chi(\vec{x},t)}{\overline{\chi_0}} := R = \int_0^\infty e^{-\frac{t'}{\tau(t')}} \cdot \frac{z}{2\sqrt{\pi K(\vec{x},t)t'^3}} \cdot e^{\left(\frac{z}{2H} - \frac{K(\vec{x},t)t'}{4H^2} - \frac{z^2}{4K(\vec{x},t)t'}\right)} \cdot dt'.
\tag{6}
$$

The ratio of measured mixing ratio and entry mixing ratio will from now on be called $R$. This is the main equation for the modified inverse method. The transport parameter $K(\vec{x},t)$ is then optimized numerically until the inverted ratio $R$ equals the

modelled or measured ratio with 5 % accuracy. The procedure is described in detail in Appx. A. In order to obtain a mathematically correct PDF, it is always enforced that $G(\vec{x},t',t)$ satisfies the following equation

$$
\int_0^\infty G(\vec{x},t',t) \cdot dt' = 1.
\tag{7}
$$

The inverse method yields the important benefit that the procedure can be applied to punctual data rather than complete vertical

profiles of mixing ratios (Ehhalt's method). This is particularly relevant for observational data, since complete vertical profiles at a fixed latitude are hard to retrieve from airborne measurements. By using the inverse method, every data point can be evaluated separately and provides information without depending on values above or below. The modified inverse method allows for a sophisticated analysis of stratospheric transport by providing a possibility to estimate age spectra from short-lived trace gases with reduced numerical effort. We tested the applicability of Ehhalt's method within the model setup of this paper

and found results that are in accordance with Ehhalt et al. (2007). As the method proves to be indeed restricted to the lowest part of the stratosphere, we decided not to consider it further on. All methods discussed here are limited to a transit-time independent transport parameter $K$ and express transport without consideration of an explicit temporal trend in the analyzed range of transit time. At the same time, resulting age spectra of both methods have the important drawback that they only show one clear peak and do not intrinsically reproduce seasonal variation in the net upward mass flux.

## 2.2 Imposed seasonal cycle

When studying seasonal variability of transport processes, it was already mentioned that the tropical upward mass flux has a distinct seasonal cycle, with a clear maximum during NH winter time and a minimum in NH summer (Rosenlof, 1995). This cycle is also visible in age spectra as recent transport model studies have shown (e.g. Reithmeier et al., 2008; Li et al., 2012a;

Ploeger and Birner, 2016). The annual mean shape of the pulse spectra is well approximated by an inverse Gaussian distribution with one obvious mode. But, age spectra for single seasons show several modes representing the variable flux of mass into stratosphere during the different seasons. This is an important feature, especially for species with a seasonal cycle (e.g. $CO_2$). The inverse method does not reproduce those multiple peaks intrinsically when inferring seasonal mean spectra, since Eq. ( 3 ) is the formulation of a monomodal inverse Gaussian PDF. The amplitude of the age spectra is likely to differ in all four

seasons as a feedback to seasonal variability in trace gas mixing ratios, but yet only monomodal.

A possible approach to derive a multimodal spectrum for a specific season is to scale the age spectrum in Eq. ( 6 ) appropriately. Rosenlof (1995) found that the strength of the tropical mass flux into the stratosphere across the tropical tropopause at 70 hPa has its maximum in NH winter and its minimum in NH summer. The strength of this upward mass transport is reflected in the

maxima and minima of the age spectrum. Knowing how the upward mass flux varies within a year might be used to adjust the age spectrum of a specific season. On the basis of the first three columns of Table 4 in Rosenlof (1995), the average ratio of the mass flux in each season relatively to the remaining three is derived, that is for instance winter relative to summer etcetera. The intrinsic monomodal spectrum of each season will then be adjusted using these flux ratios at every point in transit time. When considering, for example, an arbitrary monomodal summer spectrum, its amplitude has to be modulated by

approximately +25 % for transit times that correspond to spring and fall and by circa +60 % for transit times that represent winter according to the flux ratio. For every transit time in-between a proper intermediate ratio has to be applied. Finally, no scaling takes place at transit times that correspond to the respective season of the spectrum (summer in the example – see down below for further detail). All this can be realized mathematically by a cosine, which is phase-shifted to fit for a specific season. That results in the definition of a transit-time-dependent scaling factor $S$ for each season

$$
S(t') = \left( A + B \cdot \cos\left( \frac{2\pi}{365\,d} \cdot t' + C \right) \right). \tag{8}
$$

Herein, $A, B$ and $C$ denote constant factors that are only depending on the considered season. Due to the fact that increasing transit time is equivalent to going back in time, NH winter (December-January-February or DJF) is followed by NH fall (September-October-November or SON), then NH summer (June-July-August or JJA) and finally NH spring (March-April-

May or MAM). In order to estimate the uncertainty of the tropical upward mass flux in Rosenlof (1995), the dataset is compared to data of the ERA-Interim reanalysis (Dee et al., 2011), the MERRA-2 reanalysis (Gelaro et al., 2017), the JRA-55 reanalysis

(Kobayashi et al., 2015), the NCEP CFSR reanalysis (Saha et al., 2010) and also to data of the EMAC simulation. The resulting tropical upward mass fluxes at 70 hPa are shown in the top panel of Fig. 1. The relative seasonal cycle is quite similar throughout all datasets although the absolute values differ, with EMAC having the strongest mass flux and Rosenlof (1995) the weakest. EMAC is also the only case where the maximum is situated in SON rather than DJF, but only with a small difference. Since it is only of interest for Eq. ( 8 ) how the seasons scale relative to each other, all data show that this behavior appears quite similar. The scaling components $A$ and $B$ are derived in such way that they provide a robust approximation of the seasonal cycle in Rosenlof (1995). The resulting constants are depicted together with the phase shift $C$ in Tab. 1. When approximating the combined data of all reanalyses and EMAC similarly, the resulting scaling factors agree within 15 % with the scaling factor of the Rosenlof (1995) data. The scaled age spectrum $G_{seas}(\vec{x}, t', t)$ is then given by

$$G_{seas}(\vec{x}, t', t) = \frac{z}{2\sqrt{\pi K(\vec{x},t)t'^3}} \cdot e^{\left(\frac{z}{2H} - \frac{K(\vec{x},t)t'}{4H^2} - \frac{z^2}{4K(\vec{x},t)t'}\right)} \cdot \left(A + B \cdot \cos\left(\frac{2\pi}{365\,d} \cdot t' + C\right)\right) = G(\vec{x}, t', t) \cdot S(t'). \qquad (9)$$

As above, it is always ensured that the integral of the consequent spectrum $G_{seas}(\vec{x}, t', t)$ satisfies Eq. ( 7 ). The age spectrum $G_{seas}(\vec{x}, t', t)$ is manually normalized ($G_{seas}^N(\vec{x}, t', t)$) after the scaling process by

$$G_{seas}^N(\vec{x}, t', t) = \frac{G_{seas}(\vec{x}, t', t)}{\int_0^\infty G_{seas}(\vec{x}, t', t) \cdot dt'}. \qquad (10)$$

It is mandatory that $t'$ must be given in days ($d$) in Eq. ( 8 ) and Eq. ( 9 ) for a correct result. A year is approximated only by 365 days. The bottom panel of Fig. 1 illustrates the evolution of the complete scaling factor $S$ with transit time during one year. All scaling factors equal 1 for transit times 0 a and all integer multiples of 1 a (i.e. 2 a, 3 a, etcetera). In that way, the spectrum of the considered season remains unmodified within its season of origin during the scaling process and is then only altered when re-normalizing the whole spectrum uniformly. As intended, JJA has the weakest upward mass transport of the year with all other seasons being scaled up relatively, while DJF exhibits the strongest flux with a downscaling during every other season. The intermediate periods MAM and SON display a similar scaling only with the phase shift that was striven for by introducing parameter $C$. Besides, the scaling factor correctly shows its desired maximum on any curve at transit times that represent DJF, whereas the corresponding minimum appears always at summer transit times. Inserting this new formulation into the basic equation of the inverse method (Eq. ( 6 )), the modified version becomes

$$R_{seas} = \int_0^\infty e^{-\frac{t'}{\tau(t')}} \cdot G_{seas}^N(\vec{x}, t', t) \cdot dt'. \qquad (11)$$

The numerical procedure to find a matching age spectrum is described in Appx. A. Implementing both Eq. ( 11 ) and Eq. ( 6 ) it can be estimated whether the seasonal cycle distorts the result or contributes to a better approximation of the pulse age spectra with respect to seasonality. In theory, the annual mean spectrum should be equivalent in both versions of the inverse method if the imposed seasonal cycle works correctly. The lowermost stratosphere is hereby likely a critical region where

tropospheric air enters also through the extra-tropical tropopause. This mechanism violates the assumption of a single entry through the tropical tropopause, which is the basis of the inverse method. Transport through the extra-tropical tropopause exhibits also a distinct seasonal cycle especially in the northern hemisphere (Appenzeller et al., 1996), possibly leading to wrongly scaled spectra when using the above presented scaling factor. The inverse method is thus likely to produce distorted age spectra when applied in this critical region. The seasonal cycle is not included in the fitting procedure as a free parameter

but imposed based on mean values of the tropical upward transport. The inverse method is also not able to incorporate any temporal variabilities in the seasonality of the flux.

The use of model data for a general proof of concept of the method's capabilities is very suitable, since the idealized setup yields the advantage of lifetimes being independent of transit time and transport pathways. Scientific focus in this study is on

the method's potential to capture stratospheric transport with its underlying assumptions in a solely dynamical model setup. The model of choice is EMAC, being a potent and very well-performing chemistry climate model, which has been used in many studies regarding stratospheric transport and chemistry.

### 2.3 EMAC model simulations and data preparation

The ECHAM/MESSy Atmospheric Chemistry (EMAC) model is a numerical chemistry and climate simulation system that

includes sub-models describing tropospheric and middle atmosphere processes and their interaction with oceans, land and human influences (Jöckel et al., 2010). It uses the second version of the Modular Earth Submodel System (MESSy2) to link multi-institutional computer codes. The core atmospheric model is the 5th generation European Centre Hamburg general circulation model (Roeckner et al., 2006). For the present study we applied EMAC (ECHAM5 version 5.3.02, MESSy version 2.53.0) in the T42L90MA-resolution, i.e. with a spherical truncation of T42 (corresponding to a quadratic Gaussian grid of

approx. 2.8 by 2.8 degrees in latitude and longitude) with 90 vertical hybrid pressure levels up to 0.01 hPa. The free-running time-slice simulation covers a period of 20 model years and is performed without chemistry and ocean models, i.e. sea surface temperatures and radiatively active trace gases are prescribed. This is a valid approach, since the study focuses on transport of passive trace gases as well as conceptual tests of the inverse method. As lower boundary, observations of sea surface temperatures and sea ice concentrations from the Hadley Centre Sea Ice and Sea Surface Temperature dataset (HADISST) are

fixed at monthly mean values averaged from 1995 to 2004. Additionally, greenhouse gases are prescribed at the constant value of year 2000. The climatology is therefore set to the state of the year 2000 and has no temporal trend over the whole model period. Still, seasonal and inter-annual variability is present. Here, monthly and zonal average data are used and analyzed from

December of year 9 till November of year 19. All data in this range are then finally averaged with respect to seasons on the one hand and annually on the other hand over the complete period of 10 years to gain solid statistics.

In order to retrieve age spectra from the model itself, 40 chemically completely inert trace gases are included that are released
as a pulse every 3 months at the tropical surface between 12.5° N and 12.5° S, starting in January of year 1. After 10 years, every pulse tracer has been pulsed once and the cycle starts again with resetting the first tracer back to 0 in October of year 9 and pulse it again in January of year 10 etcetera. From this period on, complete age spectra that range over the full period of 10 years can be inferred by creating a map of the boundary impulse response (BIR) for every point on the model grid. This concept is first described by Haine et al. (2008) and refined for transient simulations by Ploeger and Birner (2016). An
illustration of such a map from our EMAC simulation is shown in Fig. 2, where a horizontal cut (backward in source time) through the complete map gives the age spectrum at a specific field time. This provides an age spectrum with a temporal resolution of 3 months. Finer resolutions would require further pulse tracers (120 for a resolution of 1 month) and are numerically expensive. Li et al. (2012a) presented a sophisticated simulation with 12 pulse tracers released in every month of one model year simulated over 20 years field time, which is then cyclically repeated to cover a period of 20 years source time.
This computationally efficient procedure provides seasonal age spectra with a high temporal resolution of 1 month by neglecting inter-annual variability of the spectra. The simulation presented in this paper, however, features inter-annual variability. To keep the balance between numerical costs and scientific value, a resolution of 3 months in source time is selected being the absolute minimum for an investigation of seasonality, where each season is represented by one point per year transit time. Together with the pulse tracers, a linearly increasing completely inert trace gas (clock tracer) is implemented to derive
the mean AoA as a lag time that the surface mixing ratio needs to reach an arbitrary location in the stratosphere. In general, the clock tracer mean AoA will be larger than the first moments of the intrinsic pulse tracer age spectra. This is a direct consequence of the transit time being limited to 10 years. The tail of the age spectrum is underestimated and an extension is required. The formulation by Ploeger and Birner (2016) is applied

$$
G(\vec{x}, t', t) = \begin{cases} G(\vec{x}, t', t) & \text{for } t' < 10 \, a \\ G(\vec{x}, t'_\omega, t) \cdot e^{-\frac{(t'-t'_\omega)}{\omega}} & \text{for } t' > 10 \, a \end{cases} \tag{12}
$$

to all seasonal pulse spectra to correct them for transit times up to 300 a (see Appx. A). $\omega$ is a scale time derived from an exponential fit of the age spectrum between $t'_\omega$ and 10 years transit time. $t'_\omega$ is set to five years as threshold for the fit in accordance with Ploeger and Birner (2016). The extended age spectra are then normalized to be mathematically correct.

For the application of the inverse method, a further set of 40 trace gases is included with prescribed constant lifetimes ranging from one month up to 118 months by steps of three months. This simplification allows for an investigation of the method's basic principle by eliminating the variability of chemical depletion. These trace gases are constantly released in the same

source region as the pulse tracers with a mixing ratio of 100 %. Figure 3 visualizes annual mean vertical profiles of five of these radioactive tracers at 85° N, 55° N and 10° N relative to the tropopause. The dashed line marks the level of 10 hPa, where multiple short-lived trace gases have a mixing ratio of less than 10 %. As this is critically close to the accuracy of the inversion, no reliable information will be gained by the respective tracers. 10 hPa marks the upper boundary for the analysis in this paper.

The maxima are located above the tropopause at extra-tropical latitudes, which is not intuitive as one would expect them to be within the tropopause layer. This might be caused by an inhomogeneous distribution of these trace gases in the troposphere due to their sole initialization in the tropical boundary layer in combination with cross-tropopause transport. Since their mixing ratio is lower in the extra-tropical troposphere than in the tropics (source region), mixing of these air masses into the corresponding region of the stratosphere could be the cause of the reduced burden in the extra-tropical tropopause layer.

Profiles in the southern hemisphere exhibit a qualitatively identical behavior (not shown).

A problem for the comparison of the spectra is that all pulse spectra are referred to earth's surface, whereas the inverse method uses the tropical tropopause as a reference layer. When trying to refer inverse spectra to the surface, results broaden significantly and do not match the pulse spectra. This is likely a consequence of the solution given by Hall and Plumb (1994)

that uses one-dimensional diffusion to approximate transport above the tropopause and might not be sufficient for application in the tropical troposphere. Still, to make the spectra comparable, the annually averaged mean AoA from the clock tracer at the tropical tropopause is derived for the evaluated field time period. It is considered to be the mean transit time from surface to the tropopause within the model. This quantity (~ 0.19 a) is then subtracted from all clock tracer mean AoA as well as transit times of the pulse age spectra ensuring that the first transit time value is still 0 a. Resulting negative values are omitted. The

tropopause is directly extracted from EMAC for the complete analysis with the exact definition being described in Jöckel et al. (2006). To reduce errors in spectra, mean AoA and variance when approaching the tropopause, data are omitted up to 30 K above the tropopause as this is described as a transition layer with major tropospheric influence (Hoor et al., 2004; Bönisch et al., 2009). Possible inadequacies of the inverse method and of the artificial seasonal cycle in the lowermost stratosphere (see Sect. 2.2) are incorporated by introducing a threshold of 1.5 years of clock mean AoA for the evaluation of the inverse method.

The clock tracer provides the most accurate intrinsic value of mean AoA available in the model. Above the threshold, the tropopause is reasonably far off and seasonality should, in theory, mostly be steered by the tropical upward mass flux. In the model data, the tropical region is defined as average between 12.5° N and 12.5° S to be consistent with the source region of the pulse and radioactive tracers. The entry mixing ratios for the inverse method are derived within this area as annual mean values.

Atmospheric pressure is in general not a suitable choice when investigating transport processes and especially mixing. In the stratosphere, bidirectional stirring occurs mainly parallel to isentropic surfaces, which makes potential temperature the best choice for such a study. To include an adequate description of all transport mechanisms, the local potential temperature

difference to the tropopause is calculated for every stratospheric model grid point and used as vertical coordinate $z$ for the inverse method. The scale height $H$ is then fitted properly along this introduced coordinate.

## 3 Results

The following section provides the results of the model study to evaluate the performance of the inverse method. Annual and seasonal spectra are presented for three different pressure levels (70 hPa, 10 hPa and 140 hPa) in the mid-latitudes at 55° N for pulse and inverse spectra, latter with and without the seasonal cycle. This is a major region of interest, where both branches of the BDC are present and mixing is especially enhanced due to wave breaking. Many observational campaigns focus specifically on the northern mid-latitudes with abundant trace gas measurements. Age spectra are derived in this specific region in order to deeply understand ongoing transport processes. In the second part, first and second moments of the distributions are shown to compare resulting spectra on a larger scale, both annual and seasonal. The moments are derived using Eq. ( 4 ) and Eq. ( 5 ) and constitute a suitable analytical tool, since an inverse Gaussian PDF can be well approximated by its mean and width. The width of a spectrum around its first moment is computed as square root of the spectrum's variance. There is also a brief analysis of the ratio of second to first moment ($\frac{\Delta^2}{\Gamma}$ or ratio of moments), which is used to infer mean AoA from measurements of very long-lived trace gases. The pulse and inverse spectra's performance is assessed and evaluated with respect to the results of Hall and Plumb (1994). Finally, tests of applicability to observational data are presented, which include the number of trace gases necessary for the inversion as well as an estimation of how uncertainties in the knowledge of chemical lifetimes influence resulting age spectra and might be compensated.

### 3.1 Stratospheric age of air spectra

**70 hPa**

Figure 4 shows age spectra at 70 hPa and 55° N. Since 70 hPa is the origin of the tropical upward mass flux, the seasonal cycle should in theory work best on this pressure surface. The annual mean inverse spectrum without seasonal cycle (dash-dotted line in panel **(a)**) complies well with the annual pulse spectrum (black solid line), with slightly reduced amplitude and overestimated tail. The modal age (transit time at maximum) of the inverse spectrum agrees qualitatively with the modal age of the pulse spectrum. Including the seasonal cycle into the spectrum (dashed line in panel **(a)**) does not noticeably alter the annual mean curve as it exhibits nearly identical modal age and amplitude. This is an indicator that the formalism of the seasonal cycle modifies the seasons but cancels out on annual average. The seasonal pulse spectra (solid curves in panels **(b)** to **(e)**) indicate that the seasonal cycle of tropical upward transport is important at this pressure level as multiple distinct maxima and minima are evolving with deviating modal ages. The inverse spectra without seasonal cycle (dash-dotted – panels **(b)** to **(e)**) show also diminished amplitudes for all seasons, but still approximate the pulse spectra quite reliably. Yet, the inverse spectra without seasonal cycle do not reflect transport seasonality due to their prescribed monomodal shape and a

comparison of modal ages with the multimodal seasonal pulse spectra is not useful. When including the seasonal cycle, however, the results (dashed – panels **(b)** to **(e)**) coincide remarkably with the pulse spectra, exhibiting equivalent minima and maxima at according transit times, but still with slightly enhanced tails and underestimated amplitudes (especially SON). The timing of modal ages for the different seasons agrees with the pulse spectra within 0.1 years and also the resemblance of the

amplitudes has increased. The transit time of the secondary and ternary peaks are then prescribed with one year distance and match the pulse spectra qualitatively well. It seems that the prescribed seasonal cycle constitutes a reasonable choice to describe seasonality in transport at this altitude. Also, it is evident that exclusion of $t_{off}$ in the formalism of the inverse method still leads to matching spectra, all with a similar offset. Clock mean AoA values range between 2.51 a (maximum – DJF) and 1.92 a (minimum – JJA) and are located above the threshold of 1.5 years (see Sect. 2.3).

**10 hPa**

Age spectra around the previously defined upper boundary (10 hPa) of the inverse method at 55° N are depicted in Fig. 5. At

this pressure level, the tail of the spectrum is important to fully describe transport. Annual mean inverse spectra with and without seasonal cycle (panel **(a)**) show a similar performance both with underestimated amplitude and enhanced tail with respect to the pulse spectra, but no significant differences between them. Also, the imposed cycle leads to similar modal ages for the annual mean spectra. Qualitative agreement of the annual inverse spectra with the annual mean pulse spectrum is nevertheless fairly good, although the modal age does not agree with the pulse peaks. Again, the inverse spectra derived here

show an offset without explicit inclusion of $t_{off}$, similar to that found in the pulse spectra. The inverse spectra without seasonal cycle (dash-dotted – panels **(b)** to **(e)**) exhibit only small differences between their seasonal amplitudes. Since the annual mean of inverse and pulse spectra are very similar, it is likely that the seasonal pulse spectra would show a similar behavior if smoothed. The inverse spectra without seasonal cycle show, just as in Fig. 4 for the 70 hPa level, underestimated amplitudes and slight overly pronounced tails. Again, a comparison of modal ages between the monomodal inverse and multimodal pulse

spectra is not useful. Including the seasonal cycle (dashed – panels **(b)** to **(e)**), the resulting spectra proof to be similar to the pulse spectra only with slightly enhanced peaks for transit times greater than 4 years, which is probably a consequence of the coarse resolution of the spectra or of the scaling factor overestimating the actual influence of the tropical upward mass flux. The modal ages of the inverse spectra's maxima agree with the pulse spectra within up to 0.15 years. Some minor peaks appear between one and two years of transit time in the inverse spectra, that are not found in the pulse spectra. This may be caused by

the coarse temporal resolution of the pulse spectra on the one hand. On the other hand, the minor peaks could be an artefact in the inverse spectra due to the prescribed cycle. Again, the seasonal cycle provides an improvement such that the inverse spectra concur reasonably with the pulse spectra in this region of the stratosphere in contrast to monomodal spectra. The corresponding clock mean AoA values range from 5.42 a (maximum – SON) to 5.16 a (minimum – DJF) and are clearly above the threshold of 1.5 years (see Sect. 2.3).

**140 hPa**

Finally, age spectra for the lower stratosphere on the 140 hPa level are shown in Fig. 6. The temporal resolution of 3 months in the pulse spectra is quite critical for an evaluation in the lower stratosphere. The annual mean inverse spectrum without the seasonal cycle (dash-dotted – panel **(a)**) fits the annual mean pulse spectrum (solid line) very well, exhibiting qualitatively similar amplitude and modal age. An exact comparison of the modal age is not useful, due to very small transit times and comparatively large uncertainties. As above, the tail of the inverse spectrum seems to be slightly overestimated compared to the pulse spectrum. It is evident that now all seasonal pulse age spectra (solid – panels **(b)** to **(e)**) display a very similar modal age with one clearly pronounced peak and only minor secondary maxima. This implies that seasonality is dominated in this area by a local seasonal cycle in the extra-tropical cross-tropopause transport and not by the tropical upward mass flux. Inverse spectra without seasonal cycle (dash-dotted – panels **(b)** to **(e)**) reproduce that shift between the seasons qualitatively, but yielding a much larger difference among the seasonal amplitudes, with an underestimated SON (slightly DJF) and overly pronounced MAM/JJA. The annual curve (dash-dotted – panel **(a)**) compares well with the pulse spectrum nevertheless, indicating that over- and underestimation cancel out coincidentally. Since seasonality seems to be controlled by local entrainment into the stratosphere in this region in the mid-latitudes, a comparison of spectra including the artificial seasonal cycle (dashed – panels **(b)** to **(e)**) with the pulse spectra reveals seasonal differences that are inordinately distinct and very similar to the inverse spectra without seasonal cycle. Also, all inverse spectra exhibit again a slightly overestimated tail. The extra-tropical cross-tropopause transport violates the assumption of single entry in the inverse method, which does not reproduce the pulse spectra correctly. The imposed cycle prescribes seasonality wrongly as it cannot include variability in local transport trough the tropopause. The cycle of this extra-tropical transport across the tropopause is visible in Fig. 6 of Appenzeller et al. (1996) and differs significantly from the variability in the tropical upward mass flux. Hence, a modification of the imposed seasonal cycle would be necessary to include this effect. All these findings coincide well with our defined threshold, since 140 hPa at 55° N corresponds to a clock mean AoA between 1.04 a (maximum – MAM) and 0.52 a (minimum – SON) and falls below the introduced threshold of 1.5 years (see Sect. 2.3).

**3.2 Analysis of moments**

Since the seasonal cycle leads to an improvement of the inverse spectra except for the lower stratosphere and no significant changes of the annual mean state, all following results of the inverse method will include the imposed cycle.

**3.2.1 Mean AoA**

The percentage differences of the annual pulse and inverse mean AoA to the clock tracer are given in Fig. 7. The pulse mean AoA (left) matches the clock mean AoA very well except for the lower stratosphere. This is expected for sharp peaks of the age spectrum, such as in Fig. 6, where the temporal resolution of three months might not be sufficient and leads to inaccurate mean AoA. The tail correction of the spectra as well as their sole initialization in the tropics might also contribute to these

deviations. For lower pressures, however, the approximation of the tail correction seems to be adequate with matching mean AoA values. The main differences between pulse and clock mean AoA also appear to follow the shaded area (right panel) of the 1.5 year threshold (see Sect. 2.3). If globally averaged for the pulse spectra, this results in deviations of -0.61 % above the threshold and +14.5 % below. These findings imply that the tail correction works mostly as intended and only needs to be

applied with caution around the tropopause region. The inverse mean AoA (right) is biased towards larger values, with largest positive differences at polar and tropical latitudes. Negative differences are only found in the southern mid-latitude lowermost stratosphere, which fall below the threshold of 1.5 years mean clock AoA (dashed) for the most part. This is most probably linked to inaccurate seasonal spectra, which do not cancel out on annual average. Below the threshold, a global difference of +44.3 % is derived, whereas above the threshold this deviation reduces to +13.3 %. The bias may be attributed to the prescribed

inverse Gaussian shape of the inverse spectra and is already indicated in the enhanced tails of all inverse spectra (Fig. 4 to 6). Since the annual mean is distorted towards larger mean AoA, it is likely that the seasonal fields show a similar behavior. To compare their structure and the reproduction of pulse spectra and clock tracer qualitatively, percentage changes relative to the annual mean state are evaluated.

Figure 8 shows mean AoA of the clock tracer, the pulse and inverse spectra as latitude pressure cross section. The left column gives the absolute annual average, the remaining four columns the percentage deviations of all seasons from each annual mean. The seasonal patterns of pulse spectra and clock tracer are matching very well with a few differences in the lowermost stratosphere (especially DJF and JJA). This coincides with the results of Fig. 7. The structures of mean AoA for the inverse method match the pulse spectra and clock tracer in each season qualitatively very well above the dashed area. Only the

amplitude of the seasonal changes seems to be enhanced in direct comparison. This indicates that the seasonality that could already be detected in Fig. 4 and 5 extends mostly to the global scale. A clear boundary of those seasonal patterns is visible in the lower stratosphere around the threshold of 1.5 years clock mean AoA (see in particular MAM or SON in the north). Below, the contours do not agree with the pulse spectra and clock tracer in multiple seasons and spatial regions and display an opposite sign as the pulse or clock tracer, especially in MAM and SON in the north. These differences generally emerge particularly in

the northern hemisphere and are only visible to some small extent in the southern hemisphere (e.g. DJF – polar below 100 hPa). This is again consistent with the results of Appenzeller et al. (1996). They state that the net mass flux across the northern extra-tropical tropopause is largest downward in late MAM and largest upward in SON, whereas the upward mass flux across the tropical tropopause has its maximum in DJF and minimum in JJA. This local entry cannot be described by an annual mean entry mixing ratio at the tropical tropopause and is not well represented in the inverse method. The seasonal cycles of transport

at the tropical and the extra-tropical tropopause differ distinctly, so that the imposed seasonal cycle is not appropriate and leads to distorted inverse spectra (see Fig. 6). A combination of both effects is most probably the reason for the differences of the inverse method from clock tracer and pulse spectra in the northern mid-latitude lower stratosphere in MAM and SON. In DJF and JJA, however, the strength of the extra-tropical cross-tropopause transport is vanishingly low (Appenzeller et al., 1996), so that an assumption of single entry through the tropical tropopause should be appropriate leading to changes in inverse mean

AoA that match clock tracer and pulse spectra. This seasonal cycle in extra-tropical cross-tropopause transport is much less pronounced in the southern hemisphere, being a plausible reason for a better performance of the inverse method there. The lower stratosphere proofs to be a critical region for the inverse method also on the global scale, where further improvements are necessary. The deviations of inverse mean AoA from the pulse mean AoA and clock tracer at the tropical tropopause,

however, are more likely only a result of the choice of an annual mean entry mixing ratio for the inverse method. The first moment of a distribution is an integrated measure of the complete spectrum and, as expected, the cross sections of mean AoA for the inverse method without seasonal cycle (not shown) are identical in structure and strength to the ones presented in Fig. 8. The seasonal cycle only modulates seasons of weak and strong transport properly, while keeping the average over the complete transit time period unchanged similar to the annual mean spectrum.

**3.2.2 Width**

The following analysis focuses on three fixed latitudes, since the results presented in Fig. 8 have already shown a good performance of the inverse method beyond 1.5 years clock mean AoA on both hemispheres. 85° N, 55° N and 10° N are selected, because the northern hemisphere appears to be more challenging to the method and is better covered by in situ measurement campaigns. The analysis on fixed latitudes is also advantageous as the second moment of a PDF is strongly

depending on its first moment. An interpolation on mean AoA as vertical coordinate would be required for a global cross section, which might lead to inaccuracies and errors. Figure 9 depicts the width of spectra as function of their mean AoA for pulse (top row) and inverse (bottom row) spectra. Mean AoA serves as vertical coordinate. Filled diamonds represent data of vertical levels above the threshold of 1.5 years. For a better comparison, the pulse width is also given as grey shading in the bottom row graphics. Seasonal differences of the pulse spectra widths appear marginal below mean AoA of three years at all

latitudes. The width of a spectrum at a given mean AoA is similar and independent of the chosen latitude. With increasing mean AoA, curves begin to fan out for the different seasons, especially at 85° N. The curves at 55° N and 85° N seem to be skewed towards smaller widths for larger mean AoA, indicating that the spectra become slightly tighter around their means at the upper boundary of the analyzed area. The inverse method again captures these effects quite well with similar curvatures, although the resulting spectra are systematically wider, just as presented in Fig. 4 to 6. A steady width ensues at larger mean

AoA than for the pulse spectra. When the necessary threshold of 1.5 years is applied, primarily the obvious outliers and only few possibly matching data points are omitted (unfilled diamonds) at 85° N and 55° N. That improves the over-all agreement of the inverse and pulse width. Seasonal fluctuations in the tropics are comparatively high over the complete data range. Since the entry mixing ratio is significantly important for transport in that region, the annual mean entry mixing ratio is most probably the cause of this variability. At 55° N and 85° N, however, the seasonality is of minor influence below four years of mean

AoA, just as for the pulse spectra. Beyond that level, the curves start to fan out similarly, but with a stronger signal in particular at polar regions above five years. Coherently, the difference between pulse and inverse spectra regarding the initial point of curves fanning out is approximately the mean AoA bias of +13.3 % detected in Fig. 7. It might be concluded that the inverse method reproduces also the pulse width in a qualitative manner, but biased towards larger values, equivalent to the results for

mean AoA and mid-latitude spectra. The lowermost stratosphere arises likewise as critical region with false width below the threshold. Since width is also an integrated value, exclusion of the imposed seasonal cycle gives an identical outcome.

### 3.2.3 Ratio of moments

The ratio of second to first moment $\frac{\Delta^2}{\Gamma}$ is crucial for some observational studies of mean AoA with mixing ratios of $SF_6$. Volk
et al. (1997) propose a second-order fit to describe the temporal trend of those two trace gases. They derive a relation to calculate mean AoA in the stratosphere up to 20 km altitude between 60° N and 70° S as a function of the fit parameters and the variance of the underlying age spectrum. A ratio of moments of 1.25 a with an uncertainty of 0.5 a is then chosen, according to Hall and Plumb (1994) and Waugh et al. (1997), to prescribe the dependence on the variance. This technique has since been refined and applied in further research (Engel et al., 2002; Engel et al., 2009; Bönisch et al., 2009; Engel et al., 2017) and is
still used to this day albeit with different parameterizations. However, due to developments in climate modelling during the past 24 years, the EMAC model might include more processes on a finer grid than the models of Hall and Plumb (1994) and Waugh et al. (1997). This study offers the possibility to reassess the ratio of moments in a modern climate model also with focus on possible seasonal variability. Figure 10 depicts the absolute pulse and inverse ratio of moments for all four seasons and annual mean. The pulse results (top row) differ in terms of numbers from the results presented in Fig. 9 of Hall and Plumb
(1994) as well as from the 1.25 a applied by Volk et al. (1997) below 20 km. The spatial distribution of the ratio of moments is nevertheless quite similar to Hall and Plumb (1994), with alike large areas of constant values in the lower stratosphere on both hemispheres in all seasons, but with increased vertical gradients at the same time. The seasonal cycle of transport affects the distribution of the ratio of moments slightly, with a maximum in DJF and a minimum in JJA on both hemispheres. Values in the southern hemisphere appear slightly larger in the lowermost stratosphere compared to the north. On the basis of Fig. 10,
an annual mean ratio of moments of approximately 2.0 years for the same spatial region as in Volk et al. (1997) is detected. The inverse method (bottom row) does not reproduce the ratio from the pulse spectra in any season with globally divergent structures and numbers. This is likely caused by the biased mean AoA and width presented in the previous sections. Since the shape of the annual mean ratio of moments of the pulse spectra compares well with absolute annual clock, pulse and also inverse mean AoA (Fig. 8), it seems plausible that the inverse spectra variance is more likely the driving factor of arising
deviations. The mathematical fact that variations of mean AoA always contribute squared to the second moment (Eq. ( 5 )) stresses this assumption. Again, removing the imposed seasonal cycle from the inverse method yields identical results.

The reason for the deviation from Hall and Plumb (1994) and Volk et al. (1997) may not only be coarser resolution and lesser complexity of the models compared to EMAC. The tail of the age spectrum is vital for a mathematically and physically correct
description of transport. The width of annual mean pulse and inverse spectra with tail lengths of 10 a, 20 a, 50 a and 300 a are shown as function of mean AoA at 55° N in Fig. 11. Unfilled diamonds denote data below 1.5 years clock mean AoA. The tail of the spectrum needs to be properly specified with an appropriate length. There seems to be no distinct difference for the pulse

spectra between a tail length of 50 and 300 years. Whereas in case of the inverse method there still appears some small shift towards larger mean AoA and increased width. This is expected, since the maximum transit time is selected in accordance with the integral over the spectrum (see Appx. A). When using the intrinsic tail length of 10 years, the ratio of moments of pulse and inverse spectra have similar orders of magnitude as presented by Hall and Plumb (1994). A maximum transit time

between 10 to 20 years was also used in Hall and Plumb (1994) (T. Hall, private communication (2018)). The spatial distribution of the inverse ratio of moments agrees very well with the pulse results if using a tail length of 10 years for both. Thus, a fully included tail is needed for physically precise results on the one hand. On the other hand, the width in Fig. 11 exhibits a much stronger dependence on the tail length than mean AoA, making variations of variance even more likely the primary factor for the discrepancies between inverse method and pulse spectra in Fig. 10.

### 3.3 Applicability to observational data

Results of the proof of concept presented above have shown that the inverse method in combination with the imposed seasonal cycle is in principle capable of deriving seasonal age spectra correctly except for the region below 1.5 years mean AoA. However, when it comes to real atmospheric data, application of the method becomes even more challenging. One of the most

critical factors is the chemical lifetime of the gases used in the inversion, which are strongly time- and space-dependent, tainted with seasonal and inter-annual variability and only known with limited accuracy. The inverse method in its form postulated above (Eq. ( 11 )) can include variability in lifetimes, since the lifetime $\tau$ is designed to be transit-time-dependent, although a constant lifetime is used for all tracers implemented in this study. Transit-time-dependence is a more approximate choice compared to real space dependence, since two distinct spatial pathways could in theory exhibit equal transit time but different

lifetimes along them. However, as discussed in Engel et al. (2018), a good average correlation between chemical loss and time spent in the stratosphere is expected. Extension of the method to variable chemistry could involve the ansatz of Schoeberl et al. (2000) and Schoeberl et al. (2005), who reduced the depletion of trace gas species to chemistry along average Lagrangian paths starting at the reference point to any location in the stratosphere (see Sect. 4).

A further limitation of the method when applying it to observational data, is the number of trace gases necessary for the inversion to work properly. In reality, it is impossible to find 40 trace gases with lifetimes ranging from 1 month to 118 months evenly spaced in steps of 3 months. Considering data and measured species of modern airborne research campaigns, it is more likely to find a set of 10 trace gases at most, which span a range of 10 years chemical lifetime. On this basis, three subsets of tracers with 10, 5 and 3 trace gas species are selected and shown in Tab. 2 together with their corresponding lifetimes to

investigate the effects of reducing the number of trace gases and introducing uncertainty in the knowledge of the lifetime. Three trace gases is considered to be the minimum in order to constrain different parts of the age spectrum (e.g. left flank of the peak, right flank and tail). A reduction of trace gases removes, on the one hand, redundant information about transport and

strongly diminishes the amount of data, but also increases the risk of errors during the inversion leading to wrong age spectra on the other hand. This is especially precarious if chemical depletion is spatially varying and inaccurately estimated.

To test the influence of uncertainties in assumed lifetimes, a Monte Carlo approach has been used in which chemical lifetimes
are varied pseudo-randomly. In this simulation the mean lower error margin is evaluated by giving a pseudo-randomly selected number of lifetimes in a trace gas subset a certain preset uncertainty $\epsilon_{lower}$ (e.g. -20 %). In a second similar run, the opposite error $\epsilon_{upper}$ (e.g. +20 %) is then applied to retrieve the mean upper margin. All simulations are repeated 5000 times for each of the three trace gas subsets (Tab. 2) while pseudo-randomly varying the lifetimes of some of the species used in the inversion. This setup will provide a mean deviation of the spectra for all cases where the exact amount of error-prone lifetimes is
unknown. Since the pseudo-random numbers are distributed uniformly, the simulation mean is equivalent to a case where the lifetimes of approximately half of the species in a subset exhibit an error, but pseudo-randomly chosen with equal probability. Each simulation is designed as a single sensitivity experiment, where lifetimes are varied constantly in transit time either upwards or downwards, which provides an estimation of the maximum and minimum error of an age spectrum for a preset uncertainty. Note that an underestimation of the lifetime causes the peak of the age spectrum to shift upward and to smaller
transit times, as depletion becomes faster. Errors of ±10 %, ±20 % and ±50 % have been selected for this study.. Results are shown in Fig. 12 at 55° N and 70 hPa as annual mean, but analog results are found at 10 hPa (not shown). As the main differences for the trace gas subsets occur within the first year of transit time, only this small slice is presented to make lines distinguishable. The age spectra of the subsets with 10 (top panel) and also 5 (mid panel) trace gases without error (solid blue line) are in good agreement with the spectrum of the full tracer set (black line). Modal age increases by 2.5 % (10 tracers) and
by 3.1 % (5 tracers) compared to the full set, whereas the amplitude changes by -0.03 % (10 tracers) and -2.5 % (5 tracers). The inversion results are robust even if using only a fourth and an eighth of all trace gases. In case of only three trace gases (bottom panel), the solid blue line exhibits a growing shift of amplitude (-9%) and also greater modal age (10 %). Overall agreement is still reasonable but not as good as for the other two subsets. The shift towards larger modal age and smaller amplitudes when decreasing the size of the subset is unexpected and not intuitive. Errors in the knowledge of chemical lifetime
of the trace gases affect all subsets equally, as all non-solid lines in all three panels show an equivalent behavior relative to the reference spectrum of each subset (solid blue lines). All averaged percentage uncertainties of amplitudes and modal ages are depicted in Tab. 3. An overestimation of chemical lifetime generally leads to smaller deviations of modal age and amplitude than an underestimation.

These results imply that a reduced set of trace gases with either 10 (optimum) or 5 (sufficient) species should be recommended in order to retrieve age spectra from observations. Values in Tab. 3 and also the general shape of the error spectra indicate that on average both of these subsets can also compensate a chemical lifetime uncertainty of up to ±20 % of a random number of species if small deviations of modal age (-10 % / +8 %) and amplitude (-8 % / +12 %) are tolerated. Given the assumptions that the inverse method is based on, the accuracy level of 5 % during the inversion and atmospheric variability when applied

to in-situ data, this threshold might be considered a reasonable choice. If the number of trace gases with lifetime error is decreased (increased) and kept constant, the deviation from the reference spectrum will become smaller (larger), due to more (less) compensation by the remaining species.

## 4 Summary and conclusion

This paper presents a modified version of the inverse method by Schoeberl et al. (2005) to derive stratospheric age of air spectra from radioactive trace gases with less free parameters. It introduces a formulation of an imposed seasonal cycle in the spectra to approximate seasonality in stratospheric transport. The development process always focuses on achieving the best possible compromise between accuracy and practicability for observational studies with very limited data. The resulting spectra are evaluated in comparison to EMAC pulse spectra. The inverse method is applied on a simplified set of 40 short-lived trace

gases with globally constant prescribed lifetimes as proof of concept. Resulting spectra are assessed as well as their first and second moments and the ratio between them. This comparison is conducted for annual mean state and for respective seasonal variation. Data within the transition layer, the first 30 K beyond the tropopause, are always omitted in our study as the strong tropospheric influence distorts the concept of age of air referred to the tropical tropopause. A threshold of 1.5 years clock mean AoA is introduced as region where entry through the local tropopause is not negligible. 10 hPa is selected as upper boundary,

since the mixing ratios of trace gases with lifetimes smaller than two years are critically close to the accuracy level of the inverse algorithm.

    The modified inverse spectra match the pulse spectra well both on annual and seasonal time scale with its new set of reduced fit parameters for most parts of the stratosphere. The imposed seasonal cycle improves the already well-described intrinsic

seasonality and reproduces seasonal variation in the spectra correctly. Multiple peaks of the inverse age spectra at 55° N coincide with the pulse spectra in time of appearance and amplitude during all seasons on 70 hPa and 10 hPa. The artificial seasonal cycle moreover does not change the moments of the distributions as they are integrated measures. This extends to the annual mean spectra, which are also almost unchanged when applying the seasonal cycle. Additionally, a need for an explicit temporal offset in the formulation of the inverse method is not detected. In the lower stratosphere at 55° N and 140 hPa, which

is below the threshold of 1.5 years mean AoA, the inverse spectra differ from the pulse spectra on the seasonal scale, whereas the annual mean matches coincidentally. The imposed seasonal cycle does not contribute to a better agreement of the seasonal spectra. In the northern extra-tropics at 140 hPa, entrainment of air through the local tropopause is important. This process is not considered in the assumption of single entry through the tropical tropopause and not included in the inverse method presented here. The seasonality of the local entry of air through the extra-tropical tropopause (Appenzeller et al., 1996) differs

strongly from the seasonality in tropical upward mass flux (Fig. 1). The imposed seasonal cycle leads to mismatching spectra as it does not take this seasonal cycle in local entrainment into account. Global mean AoA as well as spectra's width at 85° N, 55° N and 10° N reveal that the inverse method performs also well on the global scale reproducing the annual state as well as

seasonality for air masses with 1.5 years mean AoA. Below the threshold, similar problems in the seasonality of the inverse method as at 55° N arise globally in large areas around the extra-tropical tropopause. These mechanisms are enhanced in the northern hemisphere and only occur to some small extent in the south, which is again in accordance with Appenzeller et al. (1996). Due to its assumptions, the inverse method and the imposed seasonal cycle are likely not applicable in the lowermost

stratosphere below 1.5 years of mean AoA without further improvements. The chosen annual mean entry mixing ratio influences results only in close vicinity to the tropical tropopause. All inverse age spectra exhibit systematically enhanced tails leading to larger global mean AoA compared to pulse spectra and clock tracer and larger width compared to the pulse spectra, most probably caused by the prescribed inverse Gaussian shape.

The ratio of moments of the pulse and inverse spectra is compared to the results of Hall and Plumb (1994). The ratio of moments of the pulse spectra exceed their values by a factor of three, but show a similar spatial structure. Since the results show that the ratio of moments undergoes seasonal variability, an annual average value of 2.0 years for the lower stratosphere in both hemispheres is promoted. This is larger than the values implemented by Volk et al. (1997) (1.25 a ± 0.5 a) and Engel et al. (2002) (0.7 a). The large deviation between EMAC and Hall and Plumb (1994) is most probably a result of an

underestimated spectrum tail in Hall and Plumb (1994). A properly considered tail is required to fully describe stratospheric transport using age spectra, since the tail affects both mean AoA and width. As the ratio of moments is usually implemented for the derivation of mean AoA from observations (Volk et al., 1997; Engel et al., 2002), these findings might contribute to improved results. The inverse method is not capable of reproducing the pulse spectra's ratio of moments, likely because of the systematic errors in mean AoA and width not scaling accordingly.

With respect to observational data, a set of 40 trace gases with lifetimes ranging evenly spaced from one month to ten years cannot be found in reality. Tests with reduced numbers of trace gases show that subsets consisting of 5 (sufficient) to 10 (optimal) chemically active trace gases should be used in order to invert consistent and matching age spectra in the northern mid-latitudes. It is recommended that the species in these subsets should be selected so that their average local lifetimes along

the transport pathway cover a period of 10 years uniformly. The analysis shows moreover that errors in the assumed chemical lifetime, which affect a random number of trace gases in these subsets, can be compensated during the inversion to some degree. On average an uncertainty in the knowledge of lifetimes up to ±20 % still leads to reasonable agreement of age spectra if an amplitude and modal age deviation of approximately ±10 % is accepted. The more lifetimes are known correctly, the smaller these deviations will become due to better compensation by the correctly prescribed species. While the method's

presented formalism allows for varying lifetime in the form of transit-time-dependence, actual application to observations requires further development and analyses. Possible approaches might include chemistry along average Lagrangian paths (e.g. Schoeberl et al., 2000; Schoeberl et al., 2005) in combination with chemistry transport models. In addition, for the lower stratosphere with mean AoA below 1.5 years, a fractional approach as proposed by Andrews et al. (2001b) and Bönisch et al. (2009) for chemically inert trace gases could be applied, which splits the stratosphere into an upper and lower section and

derives an age spectrum as superposition of the sub-spectra of each section. If chemical depletion is then approximated by lifetimes representative for each compartment, this separation of the stratosphere possibly leads to more reasonable results than in the complete stratospheric case. In such a case, also different entry mixing ratios of short-lived trace gases at the tropical and extra-tropical tropopause could be prescribed, to better represent temporal and spatial variability. A limiting factor for long-term analyses is the fact that real stratospheric transport is not stationary and afflicted with a long-term temporal trend coupled with seasonal and inter-annual variation. This trend can neither be included in the transport parameter $K$ as it is transit-time-independent nor in the prescribed seasonal cycle. Despite these additional difficulties when applying the method to observational data, the basic principle of both modified inverse method and imposed seasonal cycle proved to work. Further work is required to make the method finally applicable to measurement data and possibly contribute to a deepened understanding of stratospheric transport. In particular, the method's application on more realistic trace gases should be included in future studies regarding observationally derived age spectra.

## Appendix A: Numerical implementation

Deriving the reference ratio $R_{ref}$ from observational or modelled data, $K$ will be numerically optimized until $R_{seas}$ or $R$ is equivalent to $R_{ref}$ within a range of 5 % accuracy. This is possible for a single trace gas as well as for a set of given species simultaneously. The more distinct substances with varying lifetimes are included, the more information about the underlying transport processes of the BDC are condensed into $K$. To estimate the deviation of $R$ from the reference ratio $R_{ref}$ for every trace gas, the mean percentage error (MPE) between them is being minimized and given as follows

$$\frac{1}{n} \cdot \sum_{i}^{n} \frac{R_{ref}^i - R^i}{R_{ref}^i} \overset{!}{=} 0 \, . \tag{A1}$$

The index $i$ denotes a single trace gas within a set of $n$ species. To achieve the optimization numerically, a combination of an algorithm postulated by Ridders (1979) and Newton's method is applied. This justifies also the choice of MPE as an measure of error beside the fact that it is a robust deviation estimate. Ridders' method is known to be convergent in any case, if and only if the root (MPE = 0) is correctly bracketed. This implies that there must be always one value of $K$ where MPE is less than 0 and one $K$ where MPE is greater than 0. Only then convergence for large deviations of $K$ from the solution is ensured and rapid. When approaching the solution of $K$, Newton's method is significantly faster converging and yet very stable, which is an advantage to keep the computational effort as small and efficient as possible. If the satisfaction criterion is reached, the corresponding age spectrum of the matching $K$ is returned. A problem arises when reaching upper or lower boundary regions. Both algorithms will become unstable in such regions. If the ratio is larger than 1, which is possible in close proximity to the tropopause where entrainment of tropospheric air occurs, or nearly equal to 0, no suitable $K$ will be optimized. In this case, the respective points will be omitted and treated as missing data. Other than in the original method of Schoeberl et al. (2005),

there are no trace gases with a constant mixing ratio of 1 at the tropical tropopause in the model, since in reality there will be uncertainties, seasonalities and temporal trends affecting this quantity. To include this effect and test if it influences the performance of the method critically, the annual mean mixing ratios are calculated at the tropical tropopause and utilized in respect of future observational approaches. The scale height $H$ is derived from the air density $\varrho$ at each latitude. This can also

be done with observed data as the density is frequently measured along with trace gases. A critical aspect for the calculation of age spectra as an inverse Gaussian PDF is the numerical choice of a value large enough to be equivalent to infinity. That is especially relevant for the tail of the age spectrum and hence for its variance, since only at infinity all possible values are considered. Multiple tests with the spectra have shown that a maximum transit time of 300 years with a resolution of 30 days is sufficient. With this selection, Eq. ( 7 ) settles at 0.999 and is not significantly changing anymore if the maximum is further

increased. All spectra are therefore be integrated from 0 a to 300 a.

**Data Availability**

All data can be made accessible on request to the authors.

**Competing interests**

The authors hereby declare that they do not have conflicting interests.

**Author contribution**

MH has written the manuscript, has evaluated the model data and has developed the ideas presented in this study in close collaboration with AE. FF and HG planned and performed the EMAC model simulation and have implemented and tested the method of Schoeberl et al. (2000) mentioned in Sect. 2.1. FF, HG and AE contributed to the preparation of the manuscript in many discussions.

**Acknowledgements**

This work is supported by the German Research Foundation (DFG) priority program 1294 (HALO) under the project number 316588118. The model simulations were performed on the HPC system Mistral of the German Climate Computation Center (DKRZ) supported by the German Federal Ministry of Education and Research (BMBF). The authors cordially thank Felix Plöger from Forschungszentrum Jülich and Harald Bönisch from Karlsruhe Institute of Technology (KIT) for their help and

the useful discussions regarding the research presented in this paper.

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

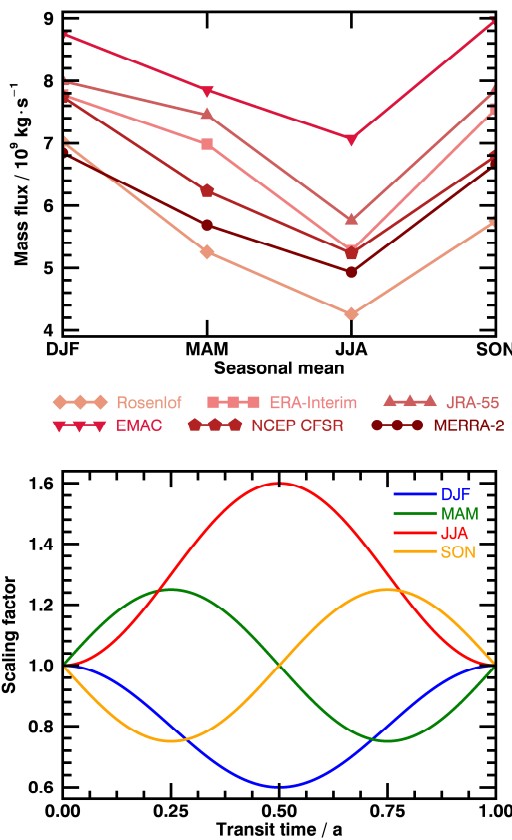

**Figure 1:** Top panel shows the tropical upward mass flux at 70 hPa. Data are from four reanalyses as well as the evaluated EMAC simulation and Rosenlof (1995). Bottom panel illustrates derived scaling factor of age spectra in one year as a function of transit time. The four seasons are color-coded. Increasing transit time means backward in time. The starting point of each curve is equivalent to the season it depicts.

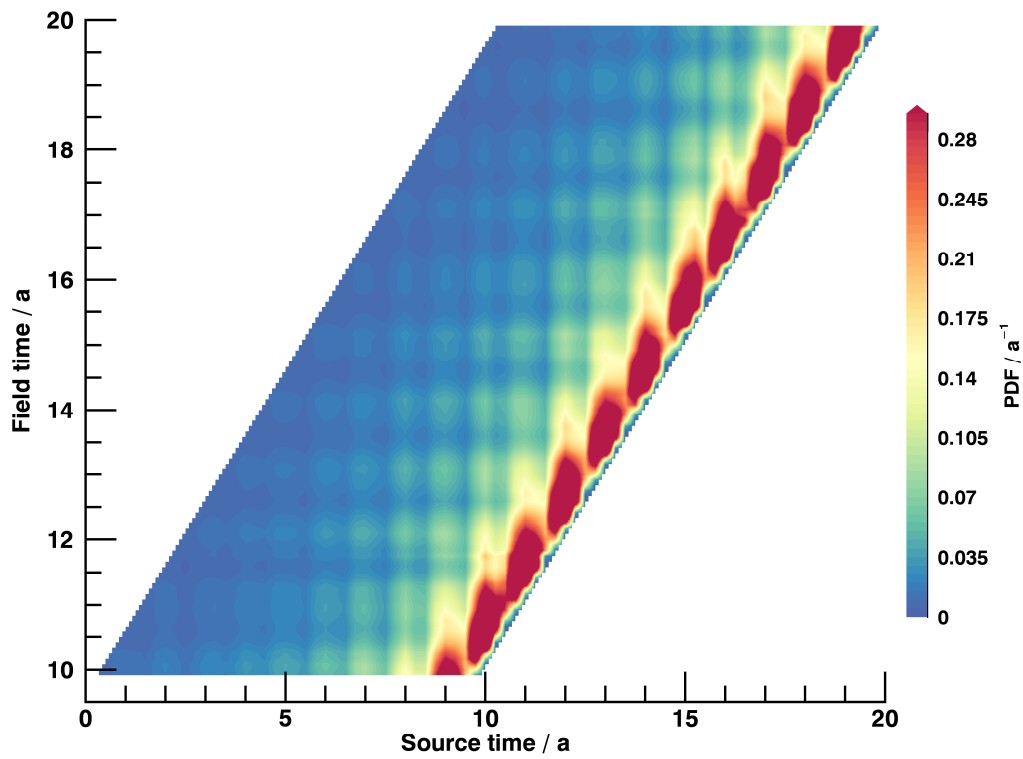

**Figure 2:** Exemplary BIR-map for the analyzed time span of the EMAC simulation at 55° N and 70 hPa. Source time represents the date when a specific pulse tracer is initialized, while field time is the progressive time of the model simulation. The amount fraction of the PDF is color-coded.

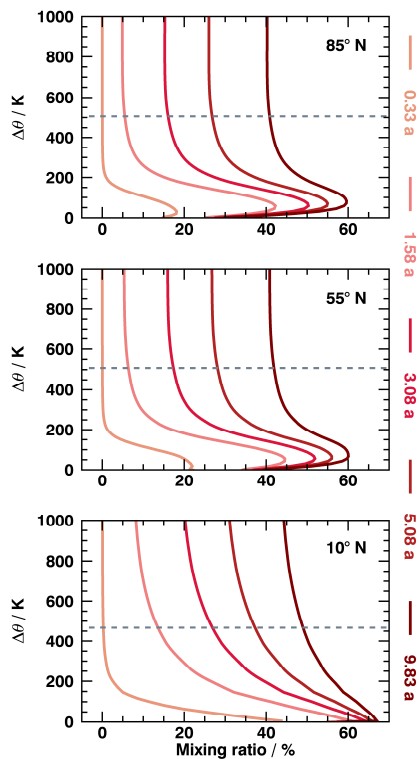

**Figure 3:** Vertical profiles of five radioactive trace gases at 85° N (**top**), 55° N (**mid**) and 10° N (**bottom**) as a function of potential temperature above tropopause. Lifetimes of species are colored. Dashed horizontal line marks the 10-hPa-level.

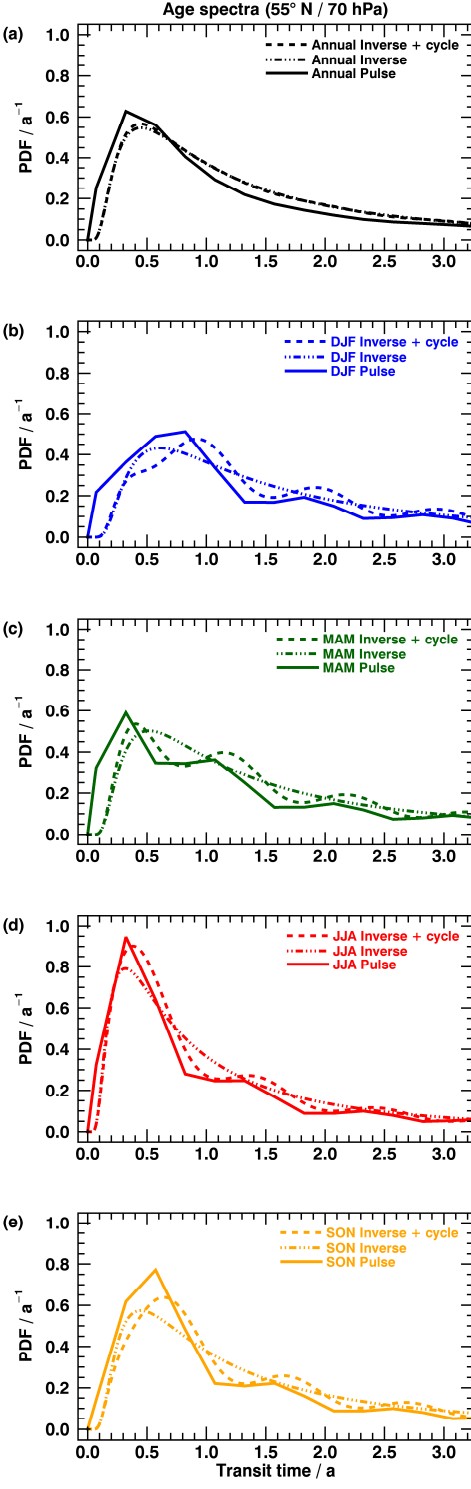

**Figure 4:** Age spectra at 55° N and 70 hPa. Panel **(a)** shows annual mean age spectra, panels **(b)** to **(e)** related seasonal spectra. Solid lines represent pulse spectra, dashed lines inverse spectra with seasonal cycle and dash-dotted lines inverse spectra without seasonal cycle.

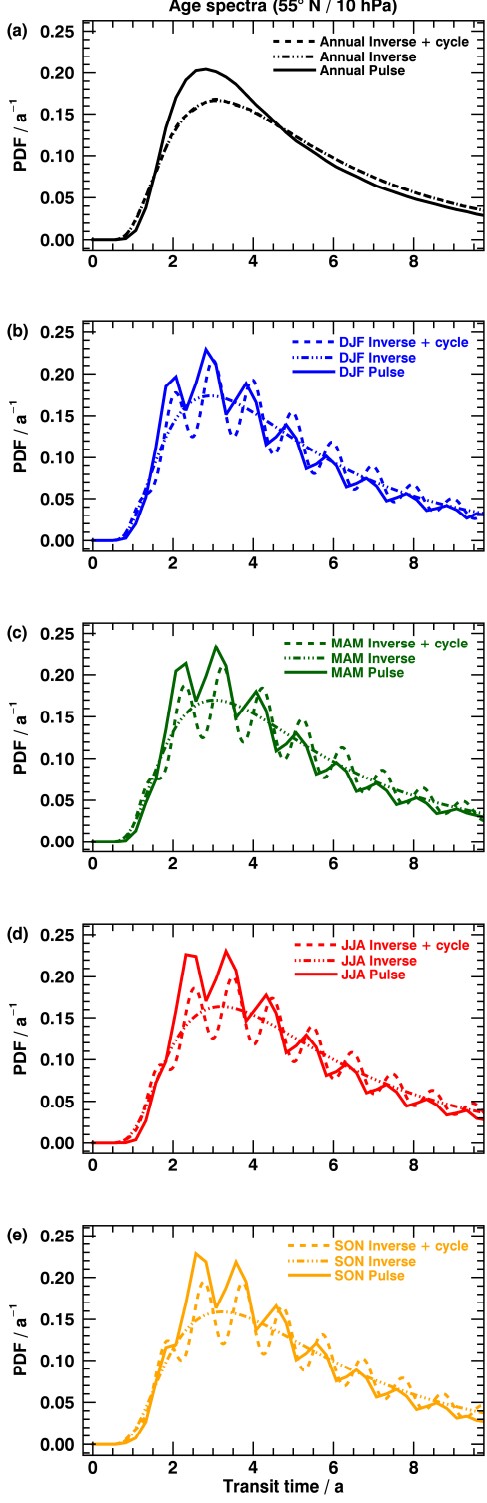

**Figure 5:** Same as Fig. 4, but at 55° N and 10 hPa.

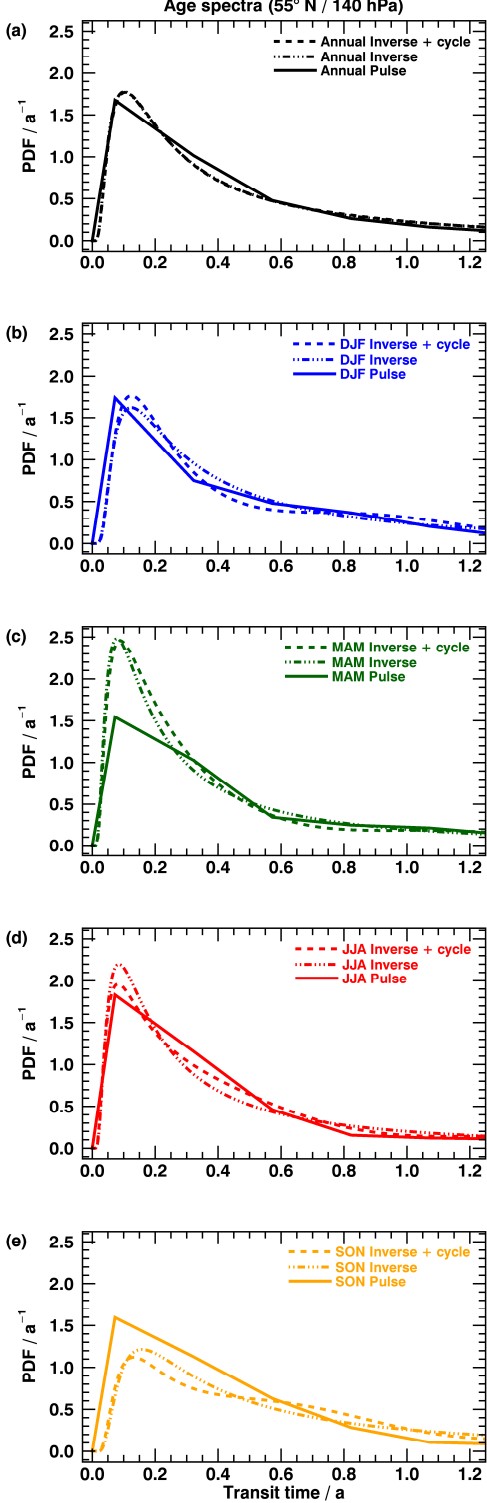

**Figure 6:** Same as Fig. 4, but at 55° N and 140 hPa.

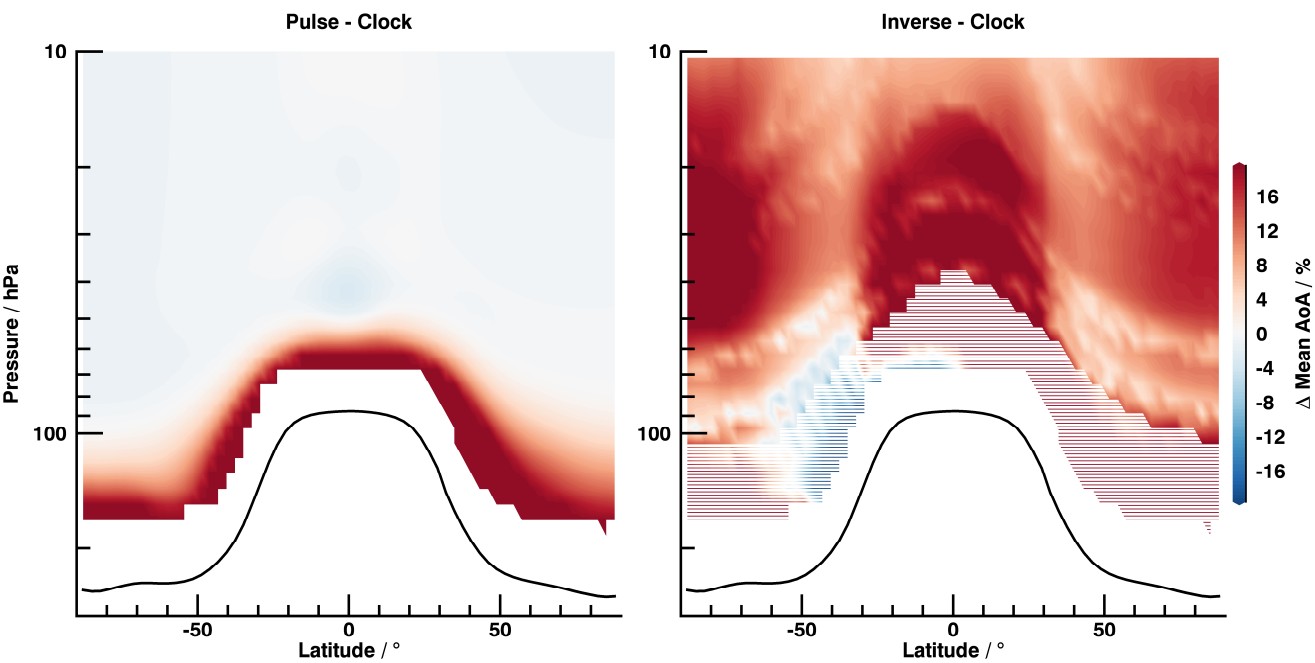

**Figure 7:** Annually averaged deviation of pulse (**left**) and inverse (**right**) mean AoA from clock mean AoA. White shaded area indicates threshold of 1.5 years clock mean AoA. Tropopause is given as solid black line. Negative x-values always denote southern hemispheric latitudes.

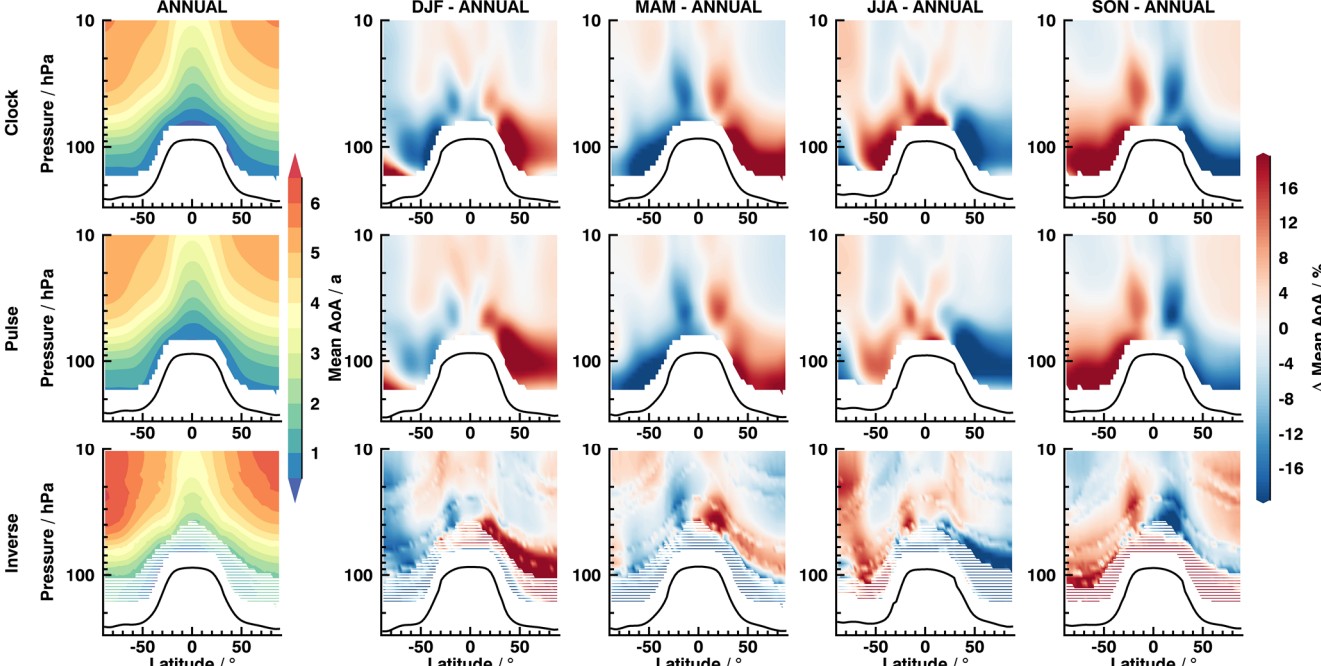

**Figure 8:** Mean AoA as global cross section. Left column shows absolute annual mean values, remaining four columns denote percentage seasonal difference to annual average. Data are from clock tracer (**Clock**), from pulse spectra (**Pulse**) and from inverse spectra including the imposed seasonal cycle (**Inverse**). The tropopause is indicated by the solid black line. White dashed area marks pressure levels below clock mean AoA of 1.5 years. The wavy structure that is visible in all seasonal plots of the inverse method, which follow hypothetical isentropic surfaces. Since potential temperature relative to the tropopause is used as vertical coordinate for the inverse method, these structures are artefacts of the implementation and do not influence the validity of results.

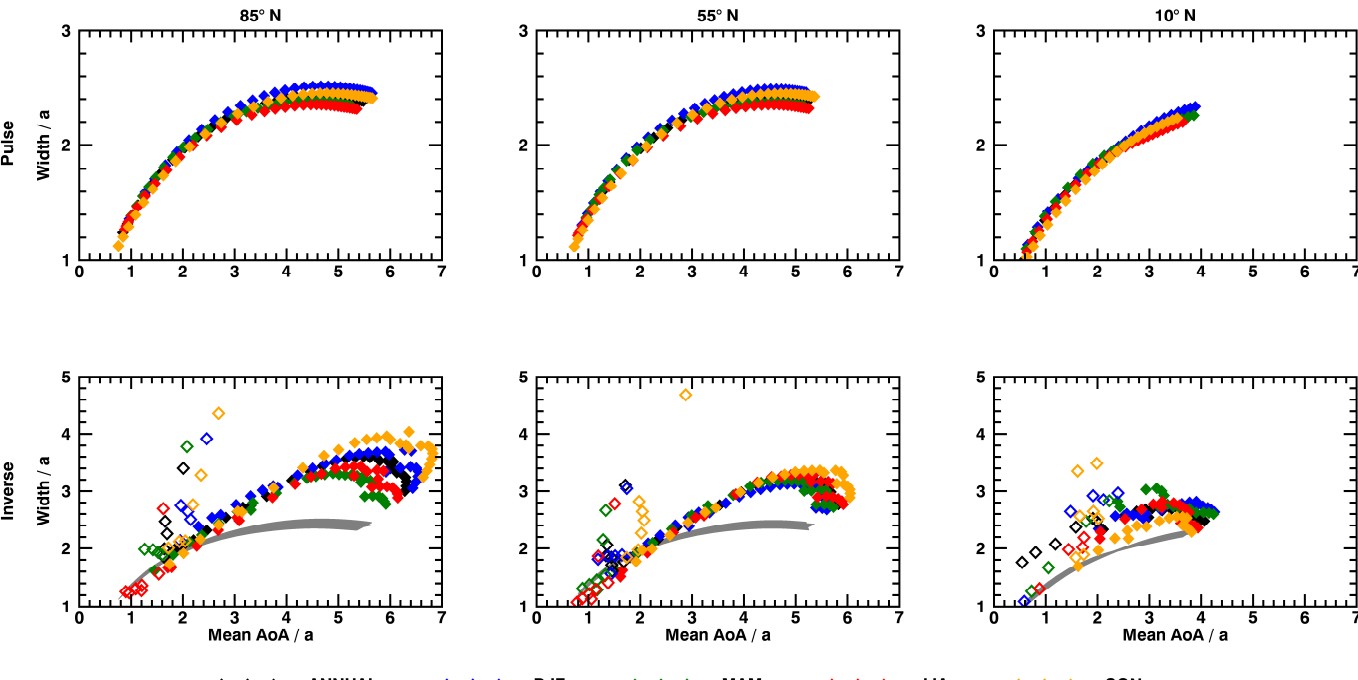

**Figure 9:** Vertical profiles of age spectra width as a function of their mean AoA at 85° N (left column), 55° N (middle column) and 10° N (right column). Top row shows pulse spectra (**Pulse**) and bottom row inverse method (**Inverse**). Seasons are given in colors. The unfilled diamonds denote data points that are found below the threshold of 1.5 years of clock mean AoA. Data of pulse spectra is also given as grey shading in bottom row for better comparison.

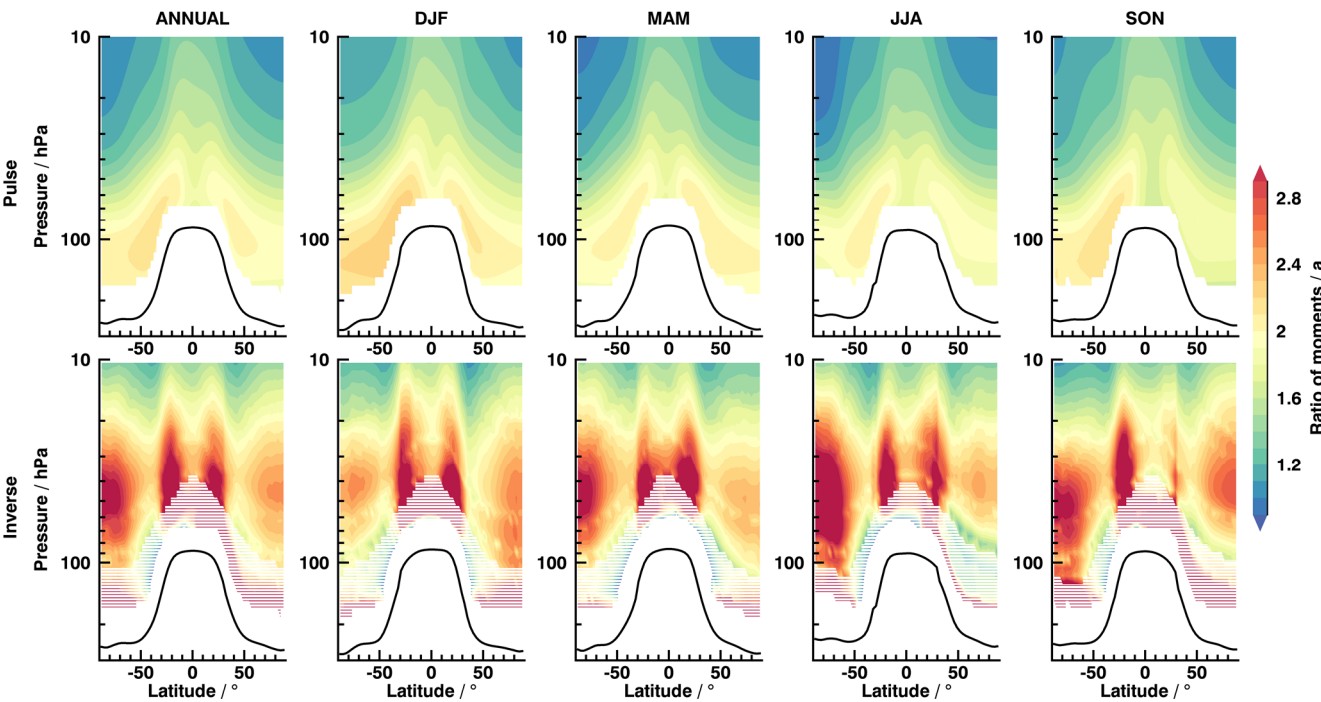

**Figure 10:** Absolute annual and seasonal ratio of moments as global cross section for pulse (**top row**) and inverse spectra (**bottom row**). Again, the white dashed area represents data below clock mean AoA of 1.5 years.

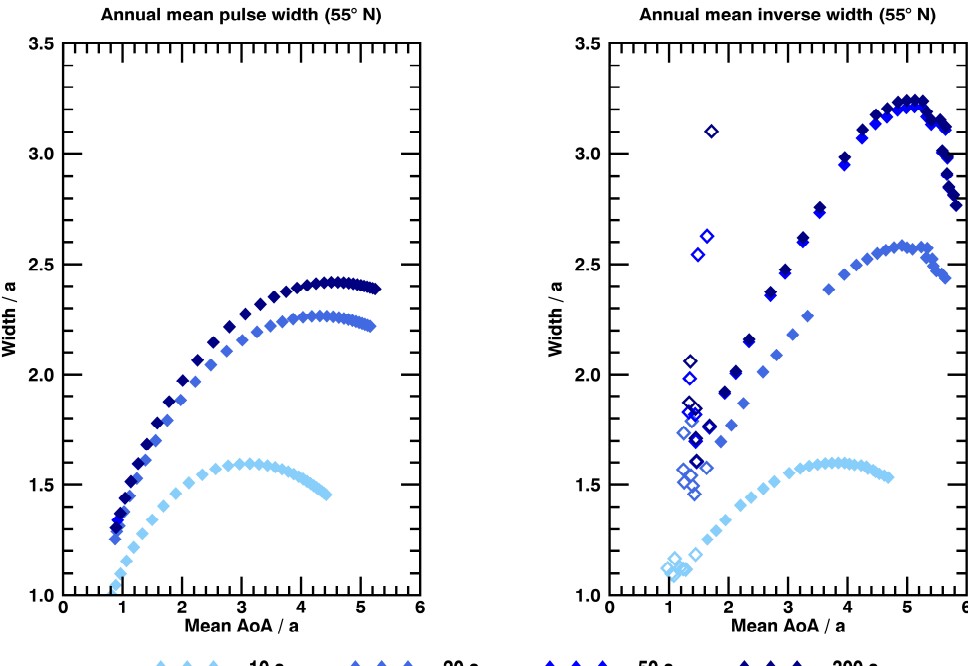

**Figure 11:** Influence of spectra's tail length on width and mean AoA for pulse (**left**) and inverse spectra (**right**) at 55° N as annual mean. Different tail lengths are colored. Filled diamonds denote data above 1.5 years of clock mean AoA.

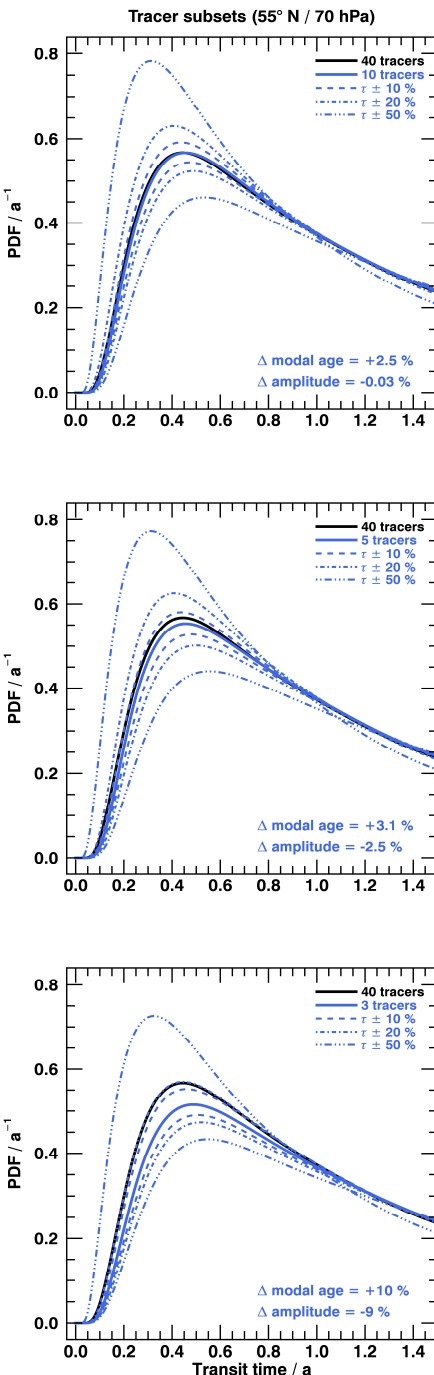

**Figure 12:** Annual mean age spectra from radioactive tracer subsets compared to full set at 55° N and 70 hPa. The black solid line corresponds to the full set (40 tracers), while the solid blue line denotes spectra of a subset with 10 tracers (**top**), 5 tracers (**mid**) and 3 tracers (**bottom**). The changes in modal age and amplitude for the solid blue lines relative to the black lines are given in each panel in the bottom right corner. The three dashed and dash-dotted spectra in each panel include an error in the knowledge of the local chemical lifetimes of ±10 %, ±20 % and ±50 % (see text for details).

**Table 1:** Scaling factors and phase shift for each season.

|  | *A* | *B* | *C* |
|---|---|---|---|
| **DJF** | 0.8 | 0.2 | $0.0 \cdot \pi$ |
| **MAM** | 1.0 | 0.25 | $1.5 \cdot \pi$ |
| **JJA** | 1.3 | 0.3 | $1.0 \cdot \pi$ |
| **SON** | 1.0 | 0.25 | $0.5 \cdot \pi$ |

5 **Table 2:** Radioactive trace gas subsets.

| Number of radioactive tracers | Selected lifetimes in months |
|---|---|
| **10** | 1, 13, 25, 37, 49, 61, 73, 85, 97 and 109 |
| **5** | 1, 25, 49, 73 and 97 |
| **3** | 1, 49 and 97 |

**Table 3:** Percentage deviations of spectra amplitudes and modal ages averaged over all three subsets presented in Fig. 12. The calculated values are relative to the spectra of each subset without errors in the knowledge of chemical lifetime.

|  | Amplitude | Modal age |
|---|---|---|
| **+10 % / -10 %** | -4.5 % / +5.5 % | +3.9 % / -4.9 % |
| **+20 % / -20 %** | -8.4 % / +11.6 % | +7.7 % / -10.2 % |
| **+50 % / -50 %** | -12.6 % / +19.4 % | +11.4 % / -16.7 % |

