# Peer review of "Deriving stratospheric age of air spectra using an idealized set of chemically active trace gases"

_Atmospheric Chemistry and Physics, 2018_

## Referee Comment (RC1) · Anonymous Referee #1 · 6 Nov 2018

This manuscript presents an inverse method to infer the stratospheric age spectra from the mixing ratio of short-lived gases. The development of a method to determine the age spectra, and not just the mean age, from observations would a major step forward in our ability to quantify stratospheric transport. Unfortunately, this manuscript does not present such a method. The method presented can estimate the age spectra from 40 tracers with spatially-uniform loss of different rates. However, these tracers don't exist, and I don't see how the method can be extended to gases with spatially varying loss. In my view for this manuscript to publishable the authors need to present a possible way for this method to be applied to real tracers. Without this I am not sure of the value of the manuscript.

MAJOR COMMENTS

1. The method is tested with a totally unrealistic situation of 40 tracers with spatially-uniform loss with loss rate varying from 1 month to 5 years. As far as I know there are no observable tracers with spatially-uniform loss let alone 40 such tracers. To test if a method will provide useful information in the real world the authors need to consider (i) a much smaller number of tracers and (ii) tracers with spatially varying loss similar to observable tracers. It would be rather straight forward to repeat the analysis using fewer than 40 tracers to see who dependent the inversion is to the number of tracers used or the range of lifetimes of these tracers, and I am very surprised the authors have not done this. But item (ii) is the more challenging and a major draw back of the proposed method. If spatially-varying loss isn't included how will this be applied to real tracers?

2. The inversion method assumes that the age spectra is an inverse Gaussian, which I think limits it value as there is a lot of evidence that this assumption breaks down in many parts of the stratosphere. With 40 tracers with different spatially-uniform loss it should be possible to estimate the age spectra without this assumption. One possible method is the maximum entropy approach (e.g. Holzer & Primeau 2010), but even simpler approaches could be possible (e.g. estimating moments of age spectra, or by assuming a more general form of the spectra such as two inversion Gaussian combined so the spectra can be bimodal). Given comment 1, this is rather academic but if going to use large number of idealized tracers than should try to estimate as much as you can about the spectra.

3. It is unfortunately that the EMAC pulse simulations were only released every 3 months, as this limits the ability to test (i) if the age spectra are inverse Gaussian and (i) the seasonality. If the aim is just to estimate the annual-mean spectra then this would not be as important, but comparisons of the shape and timing of peaks for the seasonal spectra (shown in figs 4 to 6) is really limited by the 3 month resolution.

4. There is no reference or discussion of the Li et al. 2012a,b (references below) which used the same pulse method to calculate and examine the age spectra in the GEOS

CCM. The first paper focused on the seasonal cycle and used pulses every month, and is therefore very relevant to simulations in this manuscript.

5. I think the presentation in Figures 4-6 needs to be changed. The point of these figures is to compare the spectra from the inversion with the EMAC simulated spectra, but the two spectra are shown in different plots and it is difficult to make a detailed comparison. The spectra from the inversion needs to be overlaid over the EMAC simulation so a reader can see more clearly how well the inversion works.

REFERENCES

Holzer, M., & Primeau, F. (2010). Improved constraints on transit time distributions from argon 39: A maximum entropy approach. Journal of Geophysical Research, 115, C12021. https://doi.org/10.1029/2010JC006410

Li, F., D. W. Waugh, A. R. Douglass, P. A. Newman, S. Pawson, R. S. Stolarski, S. E. Strahan, and J. E. Nielsen (2012), Seasonal variations of stratospheric age spectra in the Goddard Earth Observing System Chemistry Climate Model (GEOSCCM), J. Geophys. Res., 117, D05134, doi:10.1029/2011JD016877.

Li, F., D. W. Waugh, A. R. Douglass, P. A. Newman, S. E. Strahan, J. Ma, J. E. Nielsen, and Q. Liang (2012), Long-term changes in stratospheric age spectra in the 21st century in the Goddard Earth Observing System Chemistry-Climate Model (GEOSCCM), J. Geophys. Res., 117, D20119, doi:10.1029/2012JD017905.
* * *

---

## Referee Comment (RC2) · Anonymous Referee #3 · 7 Nov 2018

This paper uses a chemical transport model to simulate a variety of pulsed release and variable lifetime idealized trace gases to explore the seasonality of stratospheric age spectra in the model and compared to calculated age spectra based on the variable lifetime tracers that includes parameterized seasonal variability. The calculated age spectra are found by inversion from the simulated distributions of 40 trace gases with lifetimes from one to 118 months. The comparison between the pulsed tracer age spectra and the inversion-based age spectra is found to be good in most places, which seems to validate the general age spectra inversion method.

As I was reading the paper I kept waiting for the payoff of the detailed derivation and comparison of the inversion-based age spectra with the model pulse spectra. In my mind, that payoff would be an indication of the usefulness of this technique when it

comes to real stratospheric trace gas measurements. But only in the last few sentences of the Summary section is the application of the inverse age spectra method on observational data mentioned, and then only to say that it would be 'challenging'. If it actually takes measurements of 40 trace gases with lifetimes from 1 to 118 months then the application of this method would not be possible. Another potential payoff could be an improved understanding of the model transport characteristics but that seems well captured by the pulse age spectra.

It seems that more work needs to be done to help readers and the community understand the benefits and possible uses of this technique. As the paper currently stands this is not clear and although the analysis is well done it does not do enough to advance our understanding of the field in a meaningful way. The paper also has many grammatical errors, too many to list here, so it needs a more careful proofreading.

---

## Referee Comment (RC3) · Anonymous Referee #2 · 20 Nov 2018

General comment:

The paper by Hauck et al. presents a novel method to infer stratospheric age of air spectra from chemically active trace gases. This inversion method resolves seasonality in stratospheric transport and provides multimodal age spectra. As it is based on chemically active tracers it has the potential to be applied to atmospheric measurement data. Here, the method is applied and validated using a suite of trace gas species with uniform "radioactive" decay from model simulations with the EMAC model. Comparison to model pulse spectra shows very good agreement throughout most regions of the stratosphere.

The stratospheric circulation is not well constrained in current climate models, in particular regarding its variability and trends. Hence, new circulation diagnostics and con-

straints from measurements are urgently needed and the topic of the paper is highly relevant for a wide field in atmospheric science. Hence, the paper is without any doubt within the scope of ACP. The paper is well written, the results are very clearly presented and discussed within the context of existing literature in a balanced way. Problems of applying the new method, as in regions close to the tropopause (e.g., multiple stratospheric entry points) and to "real" atmospheric measurements, are critically discussed - something not all authors do so comprehensively.

Finally, I want to state clearly that I don't agree with the other Reviewer's comments which doubt the additional benefit of the paper due to being solely based on model data. Of course, deriving age spectra from "real" observations would be highly beneficial for advancing our understanding of stratospheric transport and for model validation. But this is, indeed, not a simple task. Hence, if a new method has the potential to be applied to measurement data, in my opinion the best first test is to validate the method in a controllable environment. Such a "proof of concept" is presented here based on EMAC model data, and this is also stated clearly several times (e.g., P1, L15; P4, L4).

There is clearly nothing wrong about the presented analysis. The main criticism of the other comments (applicability to measurement data) is actually already addressed and critically discussed in the discussion part. If uncertainty about such applicability to existing observations was an argument for rejecting the paper, most modelling studies would have no right to exist - and in particular many of the already published papers about stratospheric circulation and age of air. Furthermore, even if this novel method turns out not to be applicable to existing observations, with future measurements of better quality and more species it could become applicable and very relevant.

Overall, the paper is one of the better ones that I have read and I do like this validation of the age spectrum inversion method within a model environment very much. The next logical step would be the application to real measurements - but to me this is clearly a new project. Therefore, I do highly recommend publication of this study in ACP.

Only a few specific minor comments as listed below. But I leave it to the authors to decide whether they help improving the paper.

Specific comments:

P4, L17: I'm wondering whether the assumption of a constant annual mean entry mixing ratio is really necessary. If the seasonality is known it could just be included on the right-hand-side of Eq. 6 and the parameter estimation should still be possible. Maybe I miss something here - it's just a thought...

P5, L9ff: I suggest to discuss the application of the specific spectrum shape (Eq. 3) a bit more critically. The 1D diffusion approximation holds for stratospheric transport only for a stationary state global average, and on tracer coordinates (Plumb and Ko, 1992). Hence, the "diffusivity" K is not representing small-scale diffusion but including the net transport across tracer isopleths (also diffusive processes, but mainly large-scale advection).

P7, L20ff: I didn't find the explanation of the scaling factor here easy to follow and I think the explanation paragraph here could be improved. After having reached Eq. 8, and then even more the discussion of Fig. 1 (P8, L17ff), I finally understood the scaling - but it would be better to directly do so.

P10, L6: Why is the pulse reset done in October of year 9? Therefore, the spectrum is only 9 years and 9 months long. Resetting to zero in January of year ten (directly before the new pulsing) would provide exactly 10 year long spectra. I don't expect any change of the results and conlusions due to that - I'm just wondering if there was some specific argument for that...

P10, L27ff: I fully agree with the authors argumentation that the method is likely biased in the extratropical lowermost stratosphere (exLMS), due to multiple stratospheric entry points. However, the peaks in Fig. 3 above the extratropical tropopause could also be a fact and just be consistent with main entry into the exLMS via the tropical tropopause

and tropically controlled transition region in reality. The relevance of this pathway for exLMS composition has already been shown by Rosenlof et a. (1997). Further evidence, however in combination with model data, can be found in the recent study by Krause et al. (2018).

P13, L16ff: The weaker oscillations in the pulse spectra compared to the inversion results could also just be related to the coarse 3-months timing of the pulses which likely dampens the variability in the spectrum.

P19, L15: I'm wondering how critical uncertainties in the ratio of moments are for the derivation of mean age from observations. Is it possible to give a simple example here, i.e. 20% uncertainty in the ratio of moments translate into a particular uncertainty in mean age.

Technical comments:

P2, L12: Change "It" into "Mean age" P2, L23: I find "in contrast" to strong and would prefer "even". P5, L5: better say "...are then defined as (Hall and Plumb)" P7, L9: I would also cite Reithmeier et al. (2008) here for the seasonal spectrum peaks. P10, L17 (Eq. 12): There is a typo in the equation (and actually also in the same equation in Ploeger and Birner, 2016). The argument of the exponential should correctly be $t'-10a$. P20, L13: "...that it is a..."

---

## Author Comment (AC1) · 15 Jan 2019

We would like to thank all three reviewers for their constructive comments on our manuscript, which have led to major changes and significant improvements. Specifically, we have revised the following sections of the manuscript:

- The abstract has been modified to state clearly that this study is designed to provide a proof of concept of the inverse method together with its imposed seasonal cycle and check if the method can provide matching age spectra if the chemistry is well-known. We now also refer to applicability explicitly and explain the results of further tests we conducted regarding applicability of the method to real atmospheric data.

- We added a completely new paragraph in section 2.1, where we critically discuss the choice of a prescribed inverse Gaussian distribution and explain why we have decided to prescribe the general shape of the age spectrum.

- In section 2.2, we now explain the scaling factor more clearly and intuitively than before. We have also introduced a new paragraph in section 2.3, where the temporal resolution as well as the general simulation setup is discussed including the references that were mentioned by the referees.

- Finally, we added a whole new section 3.3, which discusses critical factors and problems of the method's application to observational data. In this section we also explain and present results of the newly conducted tests with the method, which include a reduction of the trace gas number as well as the method's feedback to errors in the knowledge of chemical lifetime. In this way we hope to assess the methods performance under more realistic conditions.

Changes are explained in detail below, where we answer each referee point by point. Reviewers' comments are shown in normal font, our answers in italic and changes to the manuscript in red.

**Answer to Anonymous Referee #1**

**General comment**

This manuscript presents an inverse method to infer the stratospheric age spectra from the mixing ratio of short-lived gases. The development of a method to determine the age spectra, and not just the mean age, from observations would a major step forward in our ability to quantify stratospheric transport. Unfortunately, this manuscript does not present such a method. The method presented can estimate the age spectra from 40 tracers with spatially-uniform loss of different rates. However, these tracers don't exist, and I don't see how the method can be extended to gases with spatially varying loss. In my view for this manuscript to publishable the authors need to present a possible way for this method to be applied to real tracers. Without this I am not sure of the value of the manuscript.

*We thank referee #1 for this valuable feedback. We agree that there are no 40 trace gases with constant lifetimes in reality. However, the paper is designed to be a proof of concept of the inverse methods with its modifications as well as imposed seasonal cycle. For such a study, artificial radioactive trace gases are an idealized choice, as we can assess the method's performance if chemistry is correctly prescribed. We have revised the abstract to stress this right at the beginning of the paper. The formulation of the inverse method is indeed able to include variable lifetimes, since all lifetimes can be functions of transit*

*time, but improvements in the lowermost stratosphere and final application to observational data, where lifetimes become strongly variable, require more intensive development. This is beyond the scope of this paper, but would of course be the next step with respect to applicability. We nevertheless recognize that we have to discuss critical factors and upcoming problems more critically. Therefore, section 3.3 is introduced with focus on application, where these problems are evaluated. We have tested the method's performance when using only reduced trace gas subsets and investigated how spectra change if a certain error is assumed for the description of chemical lifetimes, since it is very likely that lifetimes of real atmospheric trace gases suffer uncertainties to unknown extent (see answer to major comment 1). All this has been done under consideration of available species in past airborne research campaigns in order to be realistic. These results are also added to the abstract.*

**Major comments**

1. The method is tested with a totally unrealistic situation of 40 tracers with spatially-uniform loss with loss rate varying from 1 month to 5 years. As far as I know there are no observable tracers with spatially-uniform loss let alone 40 such tracers. To test if a method will provide useful information in the real world the authors need to consider (i) a much smaller number of tracers and (ii) tracers with spatially varying loss similar to observable tracers. It would be rather straight forward to repeat the analysis using fewer than 40 tracers to see who dependent the inversion is to the number of tracers used or the range of lifetimes of these tracers, and I am very surprised the authors have not done this. But item (ii) is the more challenging and a major draw back of the proposed method. If spatially-varying loss isn't included how will this be applied to real tracers?

*As stated above, the simulation is constructed to provide "unrealistic" chemistry intentionally. In such way, the general performance of the method and the seasonal cycle can be assessed neglecting the large variability of chemistry. This is now stated clearly in the abstract:*

"[…] This paper introduces a modified version of an inverse method to infer age spectra from mixing ratios of short-lived trace gases and investigates its basic principle in an idealized model simulation. For a full description of transport seasonality the method includes an imposed seasonal cycle to gain multimodal spectra. An EMAC model simulation is utilized for a general proof of concept of the method and features an idealized dataset of 40 radioactive trace gases with different chemical lifetimes as well as 40 chemically inert pulsed trace gases to calculate pulse age spectra. It is assessed whether the modified inverse method in combination with the seasonal cycle can provide matching age spectra when chemistry is well-known. […]"

*When it comes to real atmospheric observations, we absolutely agree with the point that there are no observable substances with completely constant depletion. However, this model setup can still be used to provide some estimation of the method's feasibility. To address problems of application more critically, we dedicate our new section 3.3 solely to this topic. We start the section by giving some evaluation of critical factors for the application:*

"[…] Results of the proof of concept presented above have shown that the inverse method in combination with the imposed seasonal cycle is in principle capable of deriving seasonal age spectra correctly except for the region below 1.5 years mean AoA. However, when it comes to real atmospheric data, application of the method becomes even more challenging. One of the most critical factors are the chemical lifetimes of the gases used in the inversion, which are strongly time- and space-dependent, tainted with seasonal and inter-annual variability and only known with limited accuracy. The inverse method in its form postulated above (Eq. ( 11 )) can include variability in lifetimes, since the lifetime $\tau$ is designed to be transit-time-dependent, although a constant lifetime is used for all tracers implemented in this study. Transit-time-dependence is a more approximative choice compared to real space dependence, since two distinct spatial pathways could in theory exhibit equal transit time but different lifetimes along them. However, as discussed in Engel et al. (2018), a good average correlation between chemical loss and time spent in the stratosphere is expected. Extension of the method to variable chemistry could involve the ansatz of Schoeberl et al. (2000) and Schoeberl et al. (2005), who reduced the depletion of trace gas species to chemistry along average Lagrangian paths starting at the reference point to any location in the stratosphere (see Sect. 4). […]"

*We have followed suggestion (i) and repeated the analysis with subsets of our radioactive trace gases consisting only of 10, 5 and 3 species with lifetimes spanning the range of 10 years. We have included this into the manuscript as follows:*

[revised manuscript text omitted]

*Finally, we changed the part about application in the conclusions (sect. 4) to match up with section 3.3. It now sums up the results of section 3.3 and provides possible approaches for future studies that aim for an extension of the inverse method:*

"[…] With respect to observational data, a set of 40 trace gases with lifetimes ranging evenly spaced from one month to ten years cannot be found in reality. Tests with reduced numbers of trace gases show that subsets consisting of 5 (sufficient) to 10 (optimal) chemically active trace gases should be used in order to invert consistent and matching age spectra in the northern mid-latitudes. It is recommended that the species in these subsets should be selected so that their average local lifetimes along the transport pathway cover a period of 10 years uniformly. The analysis shows moreover that errors in the assumed chemical lifetime, which affect a random number of trace gases in these subsets, can be compensated during the inversion to some degree. On average an uncertainty in the knowledge of lifetimes up to ±20 % still leads to reasonable agreement of age spectra if an amplitude and modal age deviation of approximately ±10 % is accepted. The more lifetimes are known correctly, the smaller these deviations will become due to better compensation by the correctly prescribed species. While the method's presented formalism allows for varying lifetime in the form of transit-time-dependence, actual application to observations requires further development and analyses. Possible approaches might include chemistry along average Lagrangian paths (e.g. Schoeberl et al., 2000; Schoeberl et al., 2005) in combination with chemistry transport models. In addition, for the lower stratosphere with mean AoA below 1.5 years, a fractional approach as proposed by Andrews et al. (2001b) and Bönisch et al. (2009) for chemically inert trace gases could be applied, which splits the stratosphere into an upper and lower section and derives an age spectrum as superposition of the sub-spectra of each section. If chemical depletion is then approximated by lifetimes representative for each compartment, this separation of the stratosphere possibly leads to more reasonable results than in the complete stratospheric case. In such a case, also different entry mixing ratios of short-lived trace gases at the tropical and extra-tropical tropopause could be prescribed, to better represent temporal and spatial variability. A limiting factor for long-term analyses is the fact that real stratospheric transport is not stationary and afflicted with a long-term temporal trend coupled with seasonal and inter-annual variation. This trend can neither be included in the transport parameter $K$ as it is transittime-independent nor in the prescribed seasonal cycle. Despite these additional difficulties when applying the method to observational data, the basic principle of both modified inverse method and imposed seasonal cycle proved to work. Further work is required to make the method finally applicable to measurement data and possibly contribute to a deepened understanding of stratospheric transport based upon observations of chemically active trace gases. […]"

2. The inversion method assumes that the age spectra is an inverse Gaussian, which I think limits it value as there is a lot of evidence that this assumption breaks down in many parts of the stratosphere. With 40 tracers with different spatially-uniform loss it should be possible to estimate the age spectra without this assumption. One possible method is the maximum entropy approach (e.g. Holzer & Primeau 2010), but even simpler approaches could be possible (e.g. estimating moments of age spectra, or by assuming a more general form of the spectra such as two inversion Gaussian combined so the spectra can be bimodal). Given comment 1, this is rather academic but if going to use large number of idealized tracers than should try to estimate as much as you can about the spectra.

*We agree with the referee and know that an inverse Gaussian shape is not a perfect match in every stratospheric region. This comment is also congruent with minor comment 2 of referee #2, which suggests to discuss the inverse Gaussian shape more critically. We always need to be aware of the problem that stratospheric data is very limited. An unconstrained shape requires more information inverted from trace gases compared to a constrained case, where all information is solely used to fit the shape as close as possible to the "real" spectrum. It is therefore not the goal to use a very large number of trace gases for the inversion, since such numbers cannot be found in reality. Many studies of age spectra from pulse experiments have shown that especially seasonal spectra are indeed multimodal, but nevertheless appear to be quite similar to an inverse Gaussian when annually averaged. This is also a reason why we implemented the seasonal cycle, to get from a rough monomodal spectrum to a much more matching multimodal shape. This is now discussed in section 2.1 and we also stress why we have decided to use the inverse Gaussian of Hall & Plumb (1994) as the basis of the inverse method:*

"[…] Transfer of Eq. ( 3 ) to complex stratospheric transport is quite difficult as Hall and Plumb (1994) state that their inverse Gaussian solution for one-dimensional diffusion is restricted to long-lived trace gases (Holton, 1986; Plumb and Ko, 1992) in a setup with only annual mean transport. In such cases, the diffusion coefficient $K$ has to be considered a measure of net vertical advection and mixing rather than simply small-scale diffusion. Their age spectra therefore do not incorporate any seasonal or inter-annual variability in transport. Model studies of annual and seasonal age spectra (Reithmeier et al., 2008; Li et al., 2012a; Li et al., 2012b; Ploeger and Birner, 2016) have shown that age spectra exhibit multiple modes representing seasonal fluctuations of stratospheric transport (see Sect. 2.2). However, even though the inverse Gaussian solution of Hall and Plumb (1994) does not include multiple peaks intrinsically and might not provide a perfect solution at any point in the stratosphere, it can certainly provide a robust approximation of the general (smoothed) shape of these multimodal spectra in

models (see e.g. Fig. 3 in Li et al. (2012a), Fig. 7 in Li et al. (2012b) or Fig. 5 in Ploeger and Birner (2016)). A prescribed age spectrum shape, even if only a fit, is indeed valuable for observational studies with very limited data. If the general shape is well constrained a priori, all information from trace gases measurements will solely be used to tune the spectrum until it is as close as possible to real transport. Approaches of this kind require less data than cases with a fully unconstrained shape, where

5    more information about transport processes is needed. Together with the relatively low amount of free parameters of the inverse Gaussian distribution in Hall and Plumb (1994), these are important factors why Eq. ( 3 ) has been the basis of many studies focusing on the retrieval of age spectra from observational data. […]"

*We also thank referee #1 for the suggested methods, especially the maximum entropy approach, which we were not aware of*
10    *before. It appears to be a very elegant approach, but we are not sure if it could be applied to limited atmospheric trace gas measurements, particularly without tracers like $^{39}Ar$ and $^{14}C$. The estimation of moments to retrieve a probability density function (i.e. a moment problem) without the PDF's moment generating or characteristic function is quite difficult. That would require to derive at least the third (skewness) or even fourth moment (kurtosis), as mean and variance do not necessarily lead to the exact shape of the underlying PDF. This would again demand larger amounts of data. A combination of two inverse*
15    *Gaussian functions is an approach that we would also consider a promising ansatz. It is basically the same approach that Andrews et al. (2001b) and Boenisch et al. (2011) considered for their study and we stress in the conclusions in section 4 that it might also work for the inverse method:*

"[…] Possible approaches might include chemistry along average Lagrangian paths (e.g. Schoeberl et al., 2000; Schoeberl et
20    al., 2005) in combination with chemistry transport models. In addition, for the lower stratosphere with mean AoA below 1.5 years, a fractional approach as proposed by Andrews et al. (2001b) and Bönisch et al. (2009) for chemically inert trace gases could be applied, which splits the stratosphere into an upper and lower section and derives an age spectrum as superposition of the sub-spectra of each section. If chemical depletion is then approximated by lifetimes representative for each compartment, this separation of the stratosphere possibly leads to more reasonable results than in the complete stratospheric case. In such a
25    case, also different entry mixing ratios of short-lived trace gases at the tropical and extra-tropical tropopause could be prescribed, to better represent temporal and spatial variability. […]"

3. It is unfortunately that the EMAC pulse simulations were only released every 3 months, as this limits the ability to test (i) if
30    the age spectra are inverse Gaussian and (i) the seasonality. If the aim is just to estimate the annual-mean spectra then this would not be as important, but comparisons of the shape and timing of peaks for the seasonal spectra (shown in figs 4 to 6) is really limited by the 3 month resolution.

*We agree with the referee that a resolution of 3 months is very coarse for a seasonal study of age spectra, especially in the lowermost stratosphere. However, since pulses are released in January, April, July and October, i.e. the middle of DJF, MAM, JJA and SON, each season is represented by exactly one pulse. We therefore believe that such a setup still enables to compare seasonal age spectra with respect to modal age and peak shape. But we also recognize that 3 months is the least possible resolution, which is now stated in the manuscript in Sect. 2.3:*

"[…] This provides an age spectrum with a temporal resolution of 3 months. Finer resolutions would require further pulse tracers (120 for a resolution of 1 month) and are numerically expensive. Li et al. (2012a) presented a sophisticated simulation with 12 pulse tracers released in every month of one model year simulated over 20 years field time, which is then cyclically repeated to cover a period of 20 years source time. This computationally efficient procedure provides seasonal age spectra with a high temporal resolution of 1 month by neglecting inter-annual variability of the spectra. The simulation presented in this paper, however, features inter-annual variability. To keep the balance between numerical costs and scientific value, a resolution of 3 months in source time is selected being the absolute minimum for an investigation of seasonality, where each season is represented by one point per year transit time. […]"

*We furthermore suggest to increase the resolution of pulses for future model studies, although this might raise computational effort significantly (60 or even 120 pulses compared to 40).*

4. There is no reference or discussion of the Li et al. 2012a,b (references below) which used the same pulse method to calculate and examine the age spectra in the GEOS CCM. The first paper focused on the seasonal cycle and used pulses every month, and is therefore very relevant to simulations in this manuscript.

*We included a reference of Li et al. (2012a, b) as those are indeed very relevant for our simulation. We discuss them shortly in multiple sections. In Sect. 2.1:*

"[…] Model studies of annual and seasonal age spectra (Reithmeier et al., 2008; Li et al., 2012a; Li et al., 2012b; Ploeger and Birner, 2016) have shown that age spectra exhibit multiple modes representing seasonal fluctuations of stratospheric transport (see Sect. 2.2). However, even though the inverse Gaussian solution of Hall and Plumb (1994) does not include multiple peaks intrinsically and might not provide a perfect solution at any point in the stratosphere, it can certainly provide a robust approximation of the general (smoothed) shape of these multimodal spectra in models (see e.g. Fig. 3 in Li et al. (2012a), Fig. 7 in Li et al. (2012b) or Fig. 5 in Ploeger and Birner (2016)). […]"

*And in Sect. 2.2:*

"[…] This cycle is also visible in age spectra as recent transport model studies have shown (e.g. Reithmeier et al., 2008; Li et al., 2012a; Ploeger and Birner, 2016). The annual mean shape of the pulse spectra is well approximated by an inverse Gaussian distribution with one obvious mode. But, age spectra for single seasons show several modes representing the variable flux of mass into stratosphere during the different seasons. […]"

*And also in Sect. 2.3:*

"[…] Li et al. (2012a) presented a sophisticated simulation with 12 pulse tracers released in every month of one model year simulated over 20 years field time, which is then cyclically repeated to cover a period of 20 years source time. This computationally efficient procedure provides seasonal age spectra with a high temporal resolution of 1 month by neglecting inter-annual variability of the spectra. […]"

5. I think the presentation in Figures 4-6 needs to be changed. The point of these figures is to compare the spectra from the inversion with the EMAC simulated spectra, but the two spectra are shown in different plots and it is difficult to make a detailed comparison. The spectra from the inversion needs to be overlaid over the EMAC simulation so a reader can see more clearly how well the inversion works.

*The layout of figure 4 – 6 has now been changed with each season and the annual mean getting a single panel. This makes comparison much easier. The corresponding section 3.1 has been adapted to match with the new figures. We also detected a minor error in our modal age data at 10 hPa. Those were the values for 70 hPa. We corrected the mistake and found that modal ages agree within 0.15 year (0.1 a before). We are sorry that this occurred, although it does not worsen the agreement. We also double-checked all other data to ensure their correctness.*

**Answer to Anonymous Referee #2**

**General comment**

The paper by Hauck et al. presents a novel method to infer stratospheric age of air spectra from chemically active trace gases. This inversion method resolves seasonality in stratospheric transport and provides multimodal age spectra. As it is based on

5   chemically active tracers it has the potential to be applied to atmospheric measurement data. Here, the method is applied and validated using a suite of trace gas species with uniform "radioactive" decay from model simulations with the EMAC model. Comparison to model pulse spectra shows very good agreement throughout most regions of the stratosphere. The stratospheric circulation is not well constrained in current climate models, in particular regarding its variability and trends. Hence, new circulation diagnostics and constraints from measurements are urgently needed and the topic of the paper is highly relevant for

10  a wide field in atmospheric science. Hence, the paper is without any doubt within the scope of ACP. The paper is well written, the results are very clearly presented and discussed within the context of existing literature in a balanced way. Problems of applying the new method, as in regions close to the tropopause (e.g., multiple stratospheric entry points) and to "real" atmospheric measurements, are critically discussed - something not all authors do so comprehensively. Finally, I want to state clearly that I don't agree with the other Reviewer's comments which doubt the additional benefit of the paper due to being

15  solely based on model data. Of course, deriving age spectra from "real" observations would be highly beneficial for advancing our understanding of stratospheric transport and for model validation. But this is, indeed, not a simple task. Hence, if a new method has the potential to be applied to measurement data, in my opinion the best first test is to validate the method in a controllable environment. Such a "proof of concept" is presented here based on EMAC model data, and this is also stated clearly several times (e.g., P1, L15; P4, L4). There is clearly nothing wrong about the presented analysis. The main criticism

20  of the other comments (applicability to measurement data) is actually already addressed and critically discussed in the discussion part. If uncertainty about such applicability to existing observations was an argument for rejecting the paper, most modelling studies would have no right to exist - and in particular many of the already published papers about stratospheric circulation and age of air. Furthermore, even if this novel method turns out not to be applicable to existing observations, with future measurements of better quality and more species it could become applicable and very relevant. Overall, the paper is one

25  of the better ones that I have read and I do like this validation of the age spectrum inversion method within a model environment very much. The next logical step would be the application to real measurements - but to me this is clearly a new project. Therefore, I do highly recommend publication of this study in ACP. Only a few specific minor comments as listed below. But I leave it to the authors to decide whether they help improving the paper.

30  *We thank referee #2 for this very positive comment on our study and very much appreciate their assessment.*

**Minor comments**

P4, L17: I'm wondering whether the assumption of a constant annual mean entry mixing ratio is really necessary. If the seasonality is known it could just be included on the right-hand-side of Eq. 6 and the parameter estimation should still be possible. Maybe I miss something here - it's just a thought...

*We agree with the referee here. In theory, the entry mixing ratio could be included in the right-hand-side of Eq. (6) and even in the integral. However, this would require a time series of the entry mixing ratio at the tropical tropopause, which is hard to attain, especially for halogenated substances. We therefore decided to simplify this problem as much as we can by trying an annual mean approach and test if we still can achieve a reasonable age spectrum. The fact that the entry mixing ratio is on*

10 *the left-hand-side of Eq. (6) is just for convenience, since we thereby normalize our mixing ratio to the tropopause. We now mention the fact that time series of short-lived substances at the tropical tropopause are hard to retrieve right above Eq. (2):*

"[…] To simplify the derivation, a constant annual mean entry mixing ratio $\overline{\chi_0} = \overline{\chi(\vec{x_0}, t - t')}$ is assumed. This keeps the balance between a physically accurate description and practicability for observational studies, since time series of short-lived

15 species at the tropical tropopause are hard to attain. Eq. ( 1 ) then becomes […]"

P5, L9ff: I suggest to discuss the application of the specific spectrum shape (Eq. 3) a bit more critically. The 1D diffusion approximation holds for stratospheric transport only for a stationary state global average, and on tracer coordinates (Plumb

20 and Ko, 1992). Hence, the "diffusivity" K is not representing small-scale diffusion but including the net transport across tracer isopleths (also diffusive processes, but mainly large-scale advection).

*This is a helpful suggestion and matches well with major comment 2 of referee #1 (please see also our answer above). We have added a discussion of the inverse Gaussian distribution including its limitations and benefits in section 2.1. There, we*

25 *also delineate why we have decided to use the inverse Gaussian formulation of Hall & Plumb (1994) for the inverse method:*

"[…] Transfer of Eq. ( 3 ) to complex stratospheric transport is quite difficult as Hall and Plumb (1994) state that their inverse Gaussian solution for one-dimensional diffusion is restricted to long-lived trace gases (Holton, 1986; Plumb and Ko, 1992) in a setup with only annual mean transport. In such cases, the diffusion coefficient $K$ has to be considered a measure of net

30 vertical advection and mixing rather than simply small-scale diffusion. Their age spectra therefore do not incorporate any seasonal or inter-annual variability in transport. Model studies of annual and seasonal age spectra (Reithmeier et al., 2008; Li et al., 2012a; Li et al., 2012b; Ploeger and Birner, 2016) have shown that age spectra exhibit multiple modes representing seasonal fluctuations of stratospheric transport (see Sect. 2.2). However, even though the inverse Gaussian solution of Hall and Plumb (1994) does not include multiple peaks intrinsically and might not provide a perfect solution at any point in the

stratosphere, it can certainly provide a robust approximation of the general (smoothed) shape of these multimodal spectra in models (see e.g. Fig. 3 in Li et al. (2012a), Fig. 7 in Li et al. (2012b) or Fig. 5 in Ploeger and Birner (2016)). A prescribed age spectrum shape, even if only a fit, is indeed valuable for observational studies with very limited data. If the general shape is well constrained a priori, all information from trace gases measurements will solely be used to tune the spectrum until it is as close as possible to real transport. Approaches of this kind require less data than cases with a fully unconstrained shape, where more information about transport processes is needed. Together with the relatively low amount of free parameters of the inverse Gaussian distribution in Hall and Plumb (1994), these are important factors why Eq. ( 3 ) has been the basis of many studies focusing on the retrieval of age spectra from observational data. […]"

P7, L20ff: I didn't find the explanation of the scaling factor here easy to follow and I think the explanation paragraph here could be improved. After having reached Eq. 8, and then even more the discussion of Fig. 1 (P8, L17ff), I finally understood the scaling - but it would be better to directly do so.

*We have adapted this specific paragraph to make the explanation clearer and included a small example to show how the scaling process works. The paragraph now reads:*

"[…] A possible approach to derive a multimodal spectrum for a specific season is to scale the age spectrum in Eq. ( 6 ) appropriately. Rosenlof (1995) found that the strength of the tropical mass flux into the stratosphere across the tropical tropopause at 70 hPa has its maximum in NH winter and its minimum in NH summer. The strength of this upward mass transport is reflected in the maxima and minima of the age spectrum. Knowing how the upward mass flux varies within a year might be used to adjust the age spectrum of a specific season. On the basis of the first three columns of Table 4 in Rosenlof (1995), the average ratio of the mass flux in each season relatively to the remaining three is derived, that is for instance winter relative to summer etcetera. The intrinsic monomodal spectrum of each season will then be adjusted using these flux ratios at every point in transit time. When considering, for example, an arbitrary monomodal summer spectrum, its amplitude has to be modulated by approximately +25 % for transit times that correspond to spring and fall and by circa +60 % for transit times that represent winter according to the flux ratio. For every transit time in-between a proper intermediate ratio has to be applied. Finally, no scaling takes place at transit times that correspond to the respective season of the spectrum (summer in the example – see down below for further detail). All this can be realized mathematically by a cosine, which is phase-shifted to fit for a specific season. That results in the definition of a transit-time-dependent scaling factor *S* for each season […]"

P10, L6: Why is the pulse reset done in October of year 9? Therefore, the spectrum is only 9 years and 9 months long. Resetting to zero in January of year ten (directly before the new pulsing) would provide exactly 10 year long spectra. I don't expect any change of the results and conlusions due to that - I'm just wondering if there was some specific argument for that...

5     *The referee is correct. Our pulse age spectra have only an intrinsic length of 9.75 years. The reset in October of year 9 is just an effect of numerical implementation in the model, that is how the pulses are counted and how the reset timer is set. We neither expect our results to change if the reset is done right before the new pulse is released.*

10 P10, L27ff: I fully agree with the authors argumentation that the method is likely biased in the extratropical lowermost stratosphere (exLMS), due to multiple stratospheric entry points. However, the peaks in Fig. 3 above the extratropical tropopause could also be a fact and just be consistent with main entry into the exLMS via the tropical tropopause and tropically controlled transition region in reality. The relevance of this pathway for exLMS composition has already been shown by Rosenlof et a. (1997). Further evidence, however in combination with model data, can be found in the recent study by Krause

15 et al. (2018).

*We definitely agree with the referee here and checked our data for such feedback. If the low mixing ratios directly above the extra-tropical tropopause were a fact and air really entered through the tropical tropopause, we would expect enhanced mean AoA in the lowermost stratosphere. This band of greater mean AoA would reach from the tropics to the extra-tropics directly*

20 *above tropopause, since transport needs to be quite slow so that there is enough time to deplete the radioactive substances to this extent. At the maximum of the vertical profile we then would expect smaller mean AoA again, i.e. faster transport. Therefore, there should be an inverted mean AoA profile in the lowermost stratosphere. However, when considering the distribution mean AoA as annual average in Fig. 8 of our manuscript, we cannot detect such a pattern in the lowermost stratosphere neither for the clock tracer nor for the pulse spectra. Mean AoA increases with altitude and no inverted pattern*

25 *is visible. Hence, we suppose that the vertical profiles of all radioactive substances are likely distorted by the initialization of the substances. This interesting fact could be tested in a future simulation by initializing all substances globally at the surface.*

P13, L16ff: The weaker oscillations in the pulse spectra compared to the inversion results could also just be related to the

30 coarse 3-months timing of the pulses which likely dampens the variability in the spectrum.

*We fully agree with the referee and added the following to the paragraph in Sect. 3.1:*

"[…], which is probably a consequence of the coarse resolution of the spectra or of the scaling factor overestimating the actual influence of the tropical upward mass flux. […]".

*We furthermore included* "[...] with slightly enhanced peaks for transit times greater than 4 years […]" *to be more precise.*

P19, L15: I'm wondering how critical uncertainties in the ratio of moments are for the derivation of mean age from observations. Is it possible to give a simple example here, i.e. 20% uncertainty in the ratio of moments translate into a particular uncertainty in mean age.

*This is an interesting aspect. Volk et al. (1997) show in their figure 5 mean AoA vs. $SF_6$ mixing ratio for a ratio of moments of 0.75 years, 1.25 years and 1.75 years. It is apparent that for greater $SF_6$, the deviation between mean AoA from all three ratios is rather small. The smaller $SF_6$ becomes, the larger this deviation grows (e.g. for 2 ppt of $SF_6$ they retrieve mean AoA between approximately 5.4 years and 5.8 years). Engel et al. (2009) have also estimated these uncertainties using a ratio of moments of 0.5 years and 1.25 years, but included it into the overall uncertainty of the data without showing them explicitly. This uncertainty of mean AoA is also strongly dependent on the nonlinearity of the $SF_6$ trend. For perfectly linear $SF_6$, the uncertainty in the ratio of moments does not influence mean AoA, as it can be determined as lag time. The more nonlinear the trend becomes, the stronger will the mean AoA depend on a chosen ratio of moments. At the end of the 1990s the temporal trend of $SF_6$ was also more nonlinear compared to recent measurements, which means that the uncertainty could have become smaller in the meantime. It might therefore be interesting to investigate this uncertainty in the envelope of future studies of $SF_6$ and mean AoA.*

**Technical comments**

P2, L12: Change "It" into "Mean age"

*We changed "it" to* "mean AoA" *to be consistent with the rest of the manuscript.*

P2, L23: I find "in contrast" to strong and would prefer "even".

*Done.*

P5, L5: better say "...are then defined as (Hall and Plumb)"

*Done.*

P7, L9: I would also cite Reithmeier et al. (2008) here for the seasonal spectrum peaks.

*Done.*

P10, L17 (Eq. 12): There is a typo in the equation (and actually also in the same equation in Ploeger and Birner, 2016). The argument of the exponential should correctly be t'-10a.

*We changed the equation to:*

10 $$"G(\vec{x}, t', t) = \begin{cases} G(\vec{x}, t', t) \ for \ t' < 10 \ a \\ G(\vec{x}, t'_\omega, t) \cdot e^{-\frac{(t'-t'_\omega)}{\omega}} \ for \ t' > 10 \ a \end{cases}"$$

*It is necessary to subtract $t'_\omega$ from $t'$, since the exponential fit starts exactly at $t'_\omega$. Therefore, the argument of the exponential function must equal 0 at $t' = t'_\omega$. Of course, the fitted spectrum is used only for $t'$ greater than 10 years. We also double-checked our code and found that it was just a typing error, while the numerical implementation has been correct all the time.*

P20, L13: "...that it is a..."

*Done.*

**Answer to Anonymous Referee #3**

**General comment**

This paper uses a chemical transport model to simulate a variety of pulsed release and variable lifetime idealized trace gases to explore the seasonality of stratospheric age spectra in the model and compared to calculated age spectra based on the variable

5  lifetime tracers that includes parameterized seasonal variability. The calculated age spectra are found by inversion from the simulated distributions of 40 trace gases with lifetimes from one to 118 months. The comparison between the pulsed tracer age spectra and the inversion-based age spectra is found to be good in most places, which seems to validate the general age spectra inversion method. As I was reading the paper I kept waiting for the payoff of the detailed derivation and comparison of the inversion-based age spectra with the model pulse spectra. In my mind, that payoff would be an indication of the

10  usefulness of this technique when it comes to real stratospheric trace gas measurements. But only in the last few sentences of the Summary section is the application of the inverse age spectra method on observational data mentioned, and then only to say that it would be 'challenging'. If it actually takes measurements of 40 trace gases with lifetimes from 1 to 118 months then the application of this method would not be possible. Another potential payoff could be an improved understanding of the model transport characteristics but that seems well captured by the pulse age spectra. It seems that more work needs to be done

15  to help readers and the community understand the benefits and possible uses of this technique. As the paper currently stands this is not clear and although the analysis is well done it does not do enough to advance our understanding of the field in a meaningful way.

*We thank referee #3 for this evaluation, which agrees well with many comments of referee #1 (especially major comment 1 of*

20  *referee #1). We also agree that a clearer outlook towards the application to observations is necessary. In order to include a payoff for the reader and to estimate the method's feasibility with regard to observational data, we have first performed a trace gas reduction. We selected subsets of species consisting of 10, 5 and 3 radioactive tracers with lifetimes spanning the complete range of 10 years. The size of those subsets of trace gases is chosen according to recent airborne research campaigns so that a set of 10, 5 and 3 species with their assumed lifetimes is more realistic than 40. This has been introduced in the new*

25  *section 3.3 that focuses solely on the method's application and discusses possible problems more critically:*

"[…] A further limitation of the method when applying it to observational data, is the number of trace gases necessary for the inversion to work properly. In reality, it is impossible to find 40 trace gases with lifetimes ranging from 1 month to 118 months evenly spaced in steps of 3 months. Considering data and measured species of modern airborne research campaigns, it is more

30  likely to find a set of 10 trace gases at most, which span a range of 10 years chemical lifetime. On this basis, three subsets of tracers with 10, 5 and 3 trace gas species are selected and shown in Tab. 2 together with their corresponding lifetimes to investigate the effects of reducing the number of trace gases and introducing uncertainty in the knowledge of the lifetime. Three trace gases is considered to be the minimum in order to constrain different parts of the age spectrum (e.g. left flank of

the peak, right flank and tail). A reduction of trace gases removes, on the one hand, redundant information about transport and strongly diminishes the amount of data, but also increases the risk of errors during the inversion leading to wrong age spectra on the other hand. This is especially precarious if chemical depletion is spatially varying and inaccurately estimated. […]"

5    *And*

"[…] Results are shown in Fig. 12 at 55° N and 70 hPa as annual mean, but analog results are found at 10 hPa (not shown). As main differences for the trace gas subsets occur within the first year of transit time, only this small slice is presented to make lines distinguishable. The age spectra of the subsets with 10 (top panel) and also 5 (mid panel) trace gases without error

10   (solid blue line) are in good agreement with the spectrum of the full tracer set (black line). Modal age increases by 2.5 % (10 tracers) and by 3.1 % (5 tracers) compared to the full set, whereas the amplitude changes by -0.03 % (10 tracers) and -2.5 % (5 tracers). The inversion results are robust even if using only a fourth and an eighth of all trace gases. In case of only three trace gases (bottom panel), the solid blue line exhibits a growing shift of amplitude (-9%) and also greater modal age (10 %). Overall agreement is still reasonable but not as good as for the other two subsets. […]"

*Another aspect with respect to applicability mainly raised by referee #1, is the spatial and temporal variability of chemical depletion in reality. Admittedly, the formulation of the inverse method is capable of handling lifetimes that are transit-time-dependent, but retrieval is not trivial and more an average approach compared to real space and time dependence. For some atmospheric trace constituents there might be a quantification of chemical lifetimes in literature, but it is always uncertain if*

20   *those values are a proper description of actual depletion. Since we cannot be certain if lifetimes are representative, we have tested how errors in the description of chemical lifetimes of some species influence the inferred age spectra. To account for the uncertainty, we have performed a Monte Carlo simulation with 5000 repetitions where we assigned certain errors (e.g. +20 %) to the knowledge of chemical lifetimes of a pseudo-randomly selected number of species in a subset of trace gases. In such way we can estimate a mean upper and lower error bound for our age spectra with respect to uncertainty in the knowledge*

25   *of chemical loss. The simulation setup and its results are also shown in section 3.3:*

[revised manuscript text omitted]

The paper also has many grammatical errors, too many to list here, so it needs a more careful proofreading.

25   *We have carefully proofread the complete paper and corrected all grammatical and typing errors we found.*

On behalf of all authors

Marius Hauck

15.01.2019

**References in our answers:**

[revised manuscript text omitted]

**Age spectra (55° N / 140 hPa)**

(a)
Annual Inverse + cycle
Annual Inverse
Annual Pulse

(b)
DJF Inverse + cycle
DJF Inverse
DJF Pulse

(c)
MAM Inverse + cycle
MAM Inverse
MAM Pulse

(d)
JJA Inverse + cycle
JJA Inverse
JJA Pulse

(e)
SON Inverse + cycle
SON Inverse
SON Pulse

Transit time / a

**Figure 6:** Same as Fig. 4, but at 55° N and 140 hPa.

[Figure]

**Figure 7:** Annually averaged deviation of pulse (**left**) and inverse (**right**) mean AoA from clock mean AoA. White shaded area indicates threshold of 1.5 years clock mean AoA. Tropopause is given as solid black line. Negative x-values always denote southern hemispheric latitudes.

[Figure]

**Figure 8:** Mean AoA as global cross section. Left column shows absolute annual mean values, remaining four columns denote percentage seasonal difference to annual average. Data are from clock tracer (**Clock**), from pulse spectra (**Pulse**) and from inverse spectra including the imposed seasonal cycle (**Inverse**). The tropopause is indicated by the solid black line. White dashed area marks pressure levels below clock mean AoA of 1.5 years. The wavy structure that is visible in all seasonal plots of the inverse method, which follow hypothetical isentropic surfaces. Since potential temperature relative to the tropopause is used as vertical coordinate for the inverse method, these structures are artefacts of the implementation and do not influence the validity of results.

[Figure]

**Figure 9:** Vertical profiles of age spectra width as a function of their mean AoA at 85° N (left column), 55° N (middle column) and 10° N (right column). Top row shows pulse spectra (**Pulse**) and bottom row inverse method (**Inverse**). Seasons are given in colors. The unfilled diamonds denote data points that are found below the threshold of 1.5 years of clock mean AoA. Data of pulse spectra is also given as grey shading in bottom row for better comparison.

[Figure]

**Figure 10:** Absolute annual and seasonal ratio of moments as global cross section for pulse (**top row**) and inverse spectra (**bottom row**). Again, the white dashed area represents data below clock mean AoA of 1.5 years.

[Figure]

**Figure 11:** Influence of spectra's tail length on width and mean AoA for pulse (**left**) and inverse spectra (**right**) at 55° N as annual mean. Different tail lengths are colored. Filled diamonds denote data above 1.5 years of clock mean AoA.

[Figure]

**Figure 12:** Annual mean age spectra from radioactive tracer subsets compared to full set at 55° N and 70 hPa. Black solid line corresponds to full set (40 tracers), while solid blue line denotes spectra of subset with 10 tracers (**top**), 5 tracers (**mid**) and 3 tracers (**bottom**). The three dashed and dash-dotted spectra in each panel include an error in the knowledge chemical lifetimes of ±10 %, ±20 % and ±50 % (see text for details).

**Table 1:** Scaling factors and phase shift for each season.

|     | $A$ | $B$ | $C$ |
|-----|-----|------|-----|
| **DJF** | 0.8 | 0.2 | $0.0 \cdot \pi$ |
| **MAM** | 1.0 | 0.25 | $1.5 \cdot \pi$ |
| **JJA** | 1.3 | 0.3 | $1.0 \cdot \pi$ |
| **SON** | 1.0 | 0.25 | $0.5 \cdot \pi$ |

5 **Table 2:** Radioactive trace gas subsets.

| Number of radioactive tracers | Selected lifetimes in months |
|-----|-----|
| **10** | 1, 13, 25, 37, 49, 61, 73, 85, 97 and 109 |
| **5** | 1, 25, 49, 73 and 97 |
| **3** | 1, 49 and 97 |

**Table 3:** Percentage deviations of spectra amplitudes and modal ages averaged over all three subsets presented in Fig. 12. The calculated values are relative to the spectra of each subset without errors in the knowledge of chemical lifetime.

|     | *Amplitude* | *Modal age* |
|-----|-----|-----|
| **+10 % / -10 %** | -4.5 % / +5.5 % | +3.9 % / -4.9 % |
| **+20 % / -20 %** | -8.4 % / +11.6 % | +7.7 % / -10.2 % |
| **+50 % / -50 %** | -12.6 % / +19.4 % | +11.4 % / -16.7 % |

---

## Author Response (AR2)

We would like to thank the co-editor and all three reviewers for their re-evaluation of our manuscript and their additional comments.

Changes are explained in detail below, where we answer each referee point by point. Co-editor's and reviewers' comments are shown in normal font, our answers in italic and changes to the manuscript in red.

**Answer to Co-Editor**

I have now three re-reviews of your paper that are not consistent, ranging from "reject" to "accept as is".

Based on the overall tenor of the reviews I have decided to select "publish subject to minor revisions". I would however ask you to take all the comments of the reviewers into account, when revising your present version (including the comment

10    regarding grammatical errors).

*We thank the co-editor for his decision and considered all comments of all three referees for the re-revision of our manuscript.*

Regarding the most critical review (although I have not followed the "reject" recommendation), I suggest that you consider these comments as potentially helpful for further improvements of your paper. For example the title is still criticised (perhaps

15    you could insert the word "idealised" here?). Perhaps you could also be a bit more explicit about the meaning/interpretation of the variable K, when it is introduced in Eq. 3.

*We have changed the title to:*

"Deriving stratospheric age of air spectra using an idealized set of chemically active trace gases".

20    *We have also included a small paragraph (p. 6 – l. 32 ff.) about the variable K in Eq. (3), which clarifies its physical meaning in the context of three dimensional transport. It reads:*

"[…] Technically, calling $K(\vec{x}, t)$ a diffusion coefficient is imprecise, since it includes information about all occurring transport processes and is not depending on a specific substance. It provides an undirected measure of transport strength and will therefore be generally referred to as transport parameter. […]".

In the abstract in l. 12 you say "assumptions" -- here perhaps "idealisations" will be more appropriate

*We changed the respective sentence to clarify its statement. It now reads*

"[…] and so far primarily based on model work. […] ".

30    Finally, when using LaTeX you could use \mathrm{for} to include the word ``for" in an equation (and similar).

*Done.*

**Answer to Anonymous Referee #1**

**General comment**:

I appreciate the additional discussion and calculations the authors have added to the manuscript on the potential applicability of the method, which has improved the manuscript. However, I am still not convinced of the value of the manuscript. I am totally supportive of model-based studies that use model output for proof of concept studies, and it is not necessary for method to be applied to data. However, I think there has to be some chance that the concept (or slight modification of it) can be applied to data (or data that could be collected). I see no evidence of this.

*We thank referee #1 for their honest assessment, although we do not agree with it.*

The challenge for current approach is the reliance of tracers that are radioactive (constant loss) with a wide range of lifetimes. What possible tracers are the authors thinking about? The sensitivity calculations with less than 40 tracers is useful for the issue of limited number of tracers, but still not anywhere near realistic. What are the possible tracers with a 1 month or even 25 month stratospheric lifetime? Perhaps more importantly is the fact that trace gases that are measured are chemically active and not radioative (so title of paper is misleading), and their loss varies spatially and with time. This makes the problem a lot more complicated (and seasonality is a second order issue compared to this). The method might be extended to include time varying loss rates but I am not convinced. This is the issue that needs to tested in a proof of concept paper, i.e. can the Schoeberl et al method be extended.

*We agree with the referee that the chemical lifetime and the set of species are two crucial parts of the method with respect to future application. That is one reason why we chose to design this work as a pre-study with idealized trace gases for a proof of concept. If the method were not working in this setup, how would it ever work under more realistic conditions. Knowing that the core inversion principle performs well, more challenging aspects can be tackled. In future studies, the transition to more realistic tracers is planned, including additional uncertainties like the knowledge of the local lifetime along the transport pathway. Please also note that we are always referring to a representative local lifetime and not stratospheric lifetime. We also added a conclusive statement for future studies to sect. 4 (p. 23, l. 9 ff.), where we stress the necessity for realistic tracers*
"[…] and possibly contribute to a deepened understanding of stratospheric transport. In particular, the method's application on more realistic trace gases should be included in future studies regarding observationally derived age spectra.".

*We moreover agree with the referee that the title of the manuscript is misleading and changed it to:*
"Deriving stratospheric age of air spectra using an idealized set of chemically active trace gases".

The manuscript is technically fine and now includes some discussion of application, so it would be OK if it was published. I am just not sure what impact it will have on advancing our understanding or others in the community.

*We thank referee #1 for this evaluation.*

**Answer to Anonymous Referee #2**

**Specific comment**:

P27, L24: Update the Podglajen & Ploeger citation as ACP.

*Done.*

**Answer to Anonymous Referee #3**

**General comment:**

The authors have done a very good job of addressing my comments as well as those of the other reviewers. The addition of the section on the applicability to observational data is interesting and useful to give an idea of how this work can begin to connect more detailed observationally derived transport with model transport. I suggest publication with consideration of the minor comments below. These comments are focused on the new observational data section.

*We thank referee #3 for their positive assessment of our manuscript.*

**Specific comments:**

Pg. 19, line 12: 'critical factors is the chemical lifetime of the gases used…'

*Done.*

Pg. 19, line 15: 'approximate' instead of 'approximative'

*Done.*

Pg. 20, line 9: 'designed as a single sensitivity'

*Done.*

Pg. 20, line 12: not sure what you mean by 'two own simulations'.

*With 'two own simulations' we want to stress that we perform one Monte Carlo simulation with a specific fixed error (e.g. -20 %) and then a second one with the opposite error (e.g. +20 %). If we were to combine them together in a single run, they would cancel out when averaging over all 5000 simulations and the spectrum would show no deviation. The formulation is indeed misleading, and we decided to exclude it from the sentence, especially since we state the fact in line 9 ff. We thank the referee for this comment. The sentence now reads:*

"[…] Errors of ±10 %, ±20 % and ±50 % have been selected for this study. […]".

Pg. 20, line 13: 'As the main differences…'

*Done.*

Pg. 20: The shift to longer modal ages with fewer tracers is interesting and a significant result. Was this an expected result? Do you know why that happens? It would be nice to include some explanation or if it's not known why then to mention that as well.

*This is indeed an interesting aspect, that we did not expect. The shift to larger modal ages could be related to a change of weighting during the inversion of the subsets. That is, for instance, with only five substances, each of the tracers influences the resulting spectrum eight times stronger than in the full set. But that is just a thought and we do not know exactly why this leads to a systematic shift towards larger modal ages and smaller amplitudes. We now state this fact in sect. 3.3 (p. 20 l. 23 ff.)*

5 "[…] The shift towards larger modal age and smaller amplitudes when decreasing the size of the subset is unexpected and not intuitive. […]".

Pg. 41, Figure 12 caption: 'error in the knowledge of the local chemical lifetimes' Not sure if you mean local lifetimes or not. Also, it would useful to include the values of the modal age shifts between the solid lines in each plot. You mention it in the 10 text but I think that's one of the main conclusions so would be good to mention again here.

*Done. We always refer to a representative local chemical lifetime. We have furthermore included the shift of modal age and amplitude into fig. 12 and adapted the caption*

[revised manuscript text omitted]

---

## Author Response (AR3)

We thank the co-editor for his acceptance of our manuscript and implemented the two technical comments listed below. Co-editor's comments are shown in normal font, our answers in italic and changes to the manuscript in red.

**Answer to Co-Editor**

l. 23: is not depending --> does not depend

*Done  (p. 6 / l. 33).*

l. 28: change to: "so far is primarily based..."

10   *We decided to split the sentence into two parts and adapted the wording proposed by the co-editor. The sentences now read (p. 1 / l. 12)*

[revised manuscript text omitted]